# Phospholipid synthesis inside phospholipid membrane vesicles

Sumie Eto[1,6], Rumie Matsumura[2,6], Yasuhiro Shimane[2,6], Mai Fujimi[1], Samuel Berhanu[1,4], Takeshi Kasama[1,5] & Yutetsu Kuruma [2,3✉]

Construction of living artificial cells from genes and molecules can expand our understanding of life system and establish a new aspect of bioengineering. However, growth and division of cell membrane that are basis of cell proliferation are still difficult to reconstruct because a high-yielding phospholipid synthesis system has not been established. Here, we developed a cell-free phospholipid synthesis system that combines fatty acid synthesis and cell-free gene expression system synthesizing acyltransferases. The synthesized fatty acids were sequentially converted into phosphatidic acids by the cell-free synthesized acyltransferases. Because the system can avoid the accumulation of intermediates inhibiting lipid synthesis, sub-millimolar phospholipids could be synthesized within a single reaction mixture. We also performed phospholipid synthesis inside phospholipid membrane vesicles, which encapsulated all the components, and showed the phospholipids localized onto the mother membrane. Our approach would be a platform for the construction of self-reproducing artificial cells since the membrane can grow sustainably.

[1] Earth-Life Science Institute, Tokyo Institute of Technology, Ookayama 2-12-1, Meguroku, Tokyo 152-8550, Japan. [2] Institute for Extra-cutting-edge Science and Technology Avant-garde Research (X-star), Japan Agency for Marine-Earth Science and Technology (JAMSTEC), Natsushima-cho 2-15, Yokosuka 237-0061, Japan. [3] Precursory Research for Embryonic Science and Technology (PRESTO), Japan Science and Technology Agency (JST), Saitama 332-0012, Japan. [4] Present address: Institute of Protein Design, Department of Biochemistry, University of Washington, Seattle, WA 98195, USA. [5] Present address: HiPep Laboratories, Kyoto, Japan. [6] These authors contributed equally: Sumie Eto, Rumie Matsumura, Yasuhiro Shimane. ✉email: ykuruma@jamstec.go.jp

Attempts to reconstitute cellular functions aim to eventually build up an entire living cell from scratch[1] for understanding the basic principles of life and better mimic natural cells[2]. Several cellular functions, such as gene expression[3], genome replication[4,5], cell division[6], and energy generation[7], have been partially or completely reconstructed by assembling the responsible biomolecules. One of the goals of artificial reconstruction is to replicate the phenomenon of self-reproduction of cells. Self-reproduction can be classified into two phases: self-regeneration of the internal components and the membrane layer[8]. For the former issue, successful reconstructions of ribosomes[9,10], tRNAs[11], circular DNA[12], and aminoacyl-tRNA synthases[13] have been reported so far. Conversely, for the latter issue, nothing to date imitates the generation of growing and dividing cell membranes based on the biological system, thus we are unable to build artificial cells that can grow their own membranes. In order to construct self-reproducing artificial cells capable of Darwinian evolution, we need to discern how to grow the membrane of artificial cells in a sustainable manner.

Recently, several groups have conducted the growth and division in artificial membrane vesicles using chemically modified lipids[14–16]. The lipids contain an amine-functionalized structure that quickly transfers acyl-chain from the donor molecule to the phospholipid precursor at the vesicle membrane, leading the membrane to grow and divide. These studies have proven that cell membrane growth and division can be reproduced as a physicochemical phenomenon when lipids are synthesized at the membrane of artificial cells. Interestingly, the same principle has been found in the L-form bacteria that can proliferate through overproducing phospholipids without using cell division machinery forming the FtsZ-ring[17]. In more biological approaches, phospholipid synthesis by the enzymes, which were synthesized in a cell-free protein synthesis system (cell-free system), has been demonstrated[18–20]. These are also performed inside liposomes (or vesicles) which encapsulate the cell-free system and the genes encoding the enzymes. These studies showed that artificial cells are able to synthesize their own phospholipids if the enzymes responsible for lipid synthesis are produced internally. However, the amounts of synthesized phospholipids were not sufficient for stimulating the vesicle membrane growth.

These previous studies highlight that the difficulty in the synthesis of phospholipids with high productivity is the core of the problem of self-growing artificial cell construction. In order to make it double the surface area of a vesicle membrane with a diameter of 30 μm, for example, 1 mM of new phospholipids must be synthesized inside the vesicle (Supplementary Table 1). Furthermore, the newly synthesized phospholipids must be integrated into the membrane. Feeding fatty acids (FAs) from outside of the vesicles is possible[21], however, they need to eventually be converted to phospholipids because only FA cannot form a stable cell membrane. Therefore, the development of a high-yielding phospholipid synthesis system is indispensable as the first step to reconstructing self-reproduction in artificial cells.

In this study, we established a unique artificial cell system capable of producing phospholipids. Unlike the previous approaches supplying acyl-coenzyme A (acyl-CoA) as a FA donor, we have reconstructed a metabolic pathway for synthesizing FAs and coupled it with a phospholipid synthesis reaction. Recombinant FA-binding proteins of type II were assembled in vitro to achieve FA synthesis. This was combined with a cell-free protein synthesis system[3] where three acyltransferases are expressed (Fig. 1a). So constructed cell-free system overcame the low productivity of phospholipid synthesis by avoiding the accumulation of intermediates inhibiting phospholipid synthesis. Furthermore, we reconstructed a CoA recycling system that reuses the reaction by-product CoA to produce acetyl-CoA and malonyl-CoA, which are direct substrates of FA synthesis. Finally, all these components were encapsulated within giant vesicles to perform phospholipid synthesis inside. We believe our method can be the platform for the construction of self-growing artificial cells toward the proliferation of artificial cells.

## Results

**In vitro fatty acid synthesis system**. Toward the construction of lipid-synthesizing artificial cell, we first reconstructed an in vitro FA synthesis system by assembling eight FA-binding enzymes (FabA, FabB, FabD, FabF, FabG, FabH, FabI, and FabZ), acyl carrier protein (ACP), and thioesterase 1 (TesA) (Fig. 1b and Supplementary Fig. 1). Briefly, FabA and FabZ are 3-hydroxyacyl-ACP dehydrases (Supplementary Fig. 2). FabB, FabF, and FabH are 3-ketoacyl-ACP synthase I, II, and III, respectively. FabD is malonyl-CoA:ACP transacylase. FabG and FabI are 3-ketoacyl-ACP reductase and enoyl-ACP reductase, respectively. ACP is a carrier of the growing fatty acid chain in fatty acid biosynthesis. TesA is a thioesterase specific for fatty acid thioesters. These enzymes were individually purified from E. coli cells and mixed with substrates ($^{13}$C-acetyl-CoA and $^{13}$C-malonyl-CoA) and electron donors NAD(P)H (a mixture of NADH and NADPH) to allow FA synthesis reaction, following the previous report[22]. The types of synthesized FAs were identified and quantified by liquid chromatography-mass spectrometry (LC/MS). The data showed that saturated FAs such as C14:0 or C16:0 were obtained as a majority of the products (Fig. 1c). The total concentration of the synthesized FAs reached 150–200 μM within five minutes of the reaction, which is consistent with the previous study[22].

Next, we modified the concentration of the components to change the balance between saturated and unsaturated FAs in the products. The balance between saturated and unsaturated FAs within the structure of phospholipids is important in terms of the fluidity of the lipid membrane. FabA and FabZ share the same step of dehydration in the FA synthesis scheme (Supplementary Fig. 2), while FabA has an additional activity as isomerase. Since the ratio between these two enzymes determines the abundance of unsaturated FAs[23], the concentrations of FabA, FabB, and FabZ were changed from 1: 1: 10 to 10: 10: 1 (μM), respectively. The FabB recognizes the isomerized fatty acyl-ACP for the next cycle of elongation. As a result, the ratio of unsaturated FA increased to about 70% when FabA and FabB concentrations were high (Fig. 1d), whereas only 15% of unsaturated FAs were synthesized when the FabA was low (Fig. 1c). In both cases, the yield quickly reached a plateau within 10 min and did not increase furthermore, even when additional substrates or electron donors were supplied at the point of the plateau (Supplementary Fig. 3).

**Inhibition of fatty acid synthesis by synthesized fatty acids**. While FA synthesis, we found formation of tiny aggregates (Fig. 1e). The aggregates were precipitated by centrifugation and analyzed by SDS-PAGE, revealing that those were mostly FabZ (Fig. 1f). The aggregates did not appear when the essential enzyme for FA synthesis (e.g., ACP) was omitted (Fig. 2e, f, Supplementary Fig. 4) where FA synthesis did not occur. This implies the synthesized FAs induced the abnormal structural formation of FabZ.

Given the results of the reaction arrest and the enzyme aggregations, we thought that the synthesized FAs impede the FA synthesis system. To verify this, we tried to synthesize FA in the presence of FA (C18:1, oleic acid) at various concentrations. The yield of FA synthesis drastically decreased as the concentration of oleic acid increased; the yield reduced to 25% when more than 200 μM of oleic acid was added (Fig. 1g). Conversely, the yield increased up to 1.5-fold when 150 μg/mL of liposomes (small size

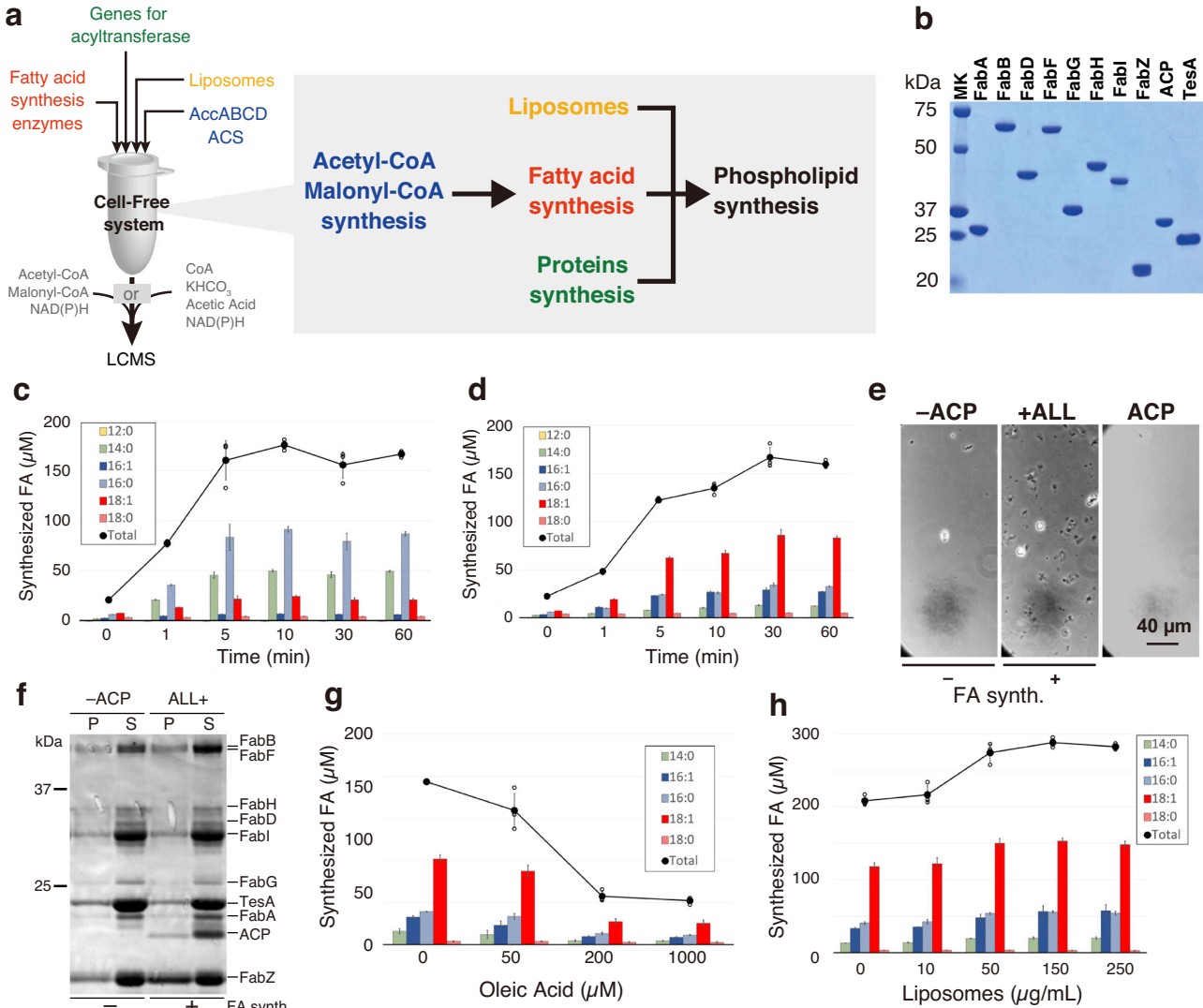

**Fig. 1 Fatty acid (FA) synthesis in a reconstructed cell-free system. a** Schematic overview of the cell-free lipid synthesis system. The system is composed of protein synthesis, acetyl-CoA and malonyl-CoA syntheses, FA synthesis, and phospholipid synthesis reactions. After the synthesis of acyltransferases on the liposome membranes, the FA synthesis is initiated by the addition of NAD(P)H and substrates, a mixture of acetyl-CoA and malonyl-CoA. When ACS and AccABCD are supplied, CoA, $KHCO_3$, and acetic acid are used as substrates to synthesize acetyl-CoA and malonyl-CoA. The synthesized FAs are converted into phospholipids by the cell-free synthesized acyltransferases. **b** The purified Fab enzymes, ACP, and TesA. **c** FA synthesis in the reconstituted in vitro system containing 10 μM FabZ, 1 μM FabA, and 1 μM FabB or **d** 1 μM FabZ, 10 μM FabA and 10 μM FabB. The types of synthesized FAs and the total yield are shown in the inset. **e** Optical microscopy images of the reaction mixture after FA synthesis. +All and −ACP indicate the mixture containing all enzymes or missing ACP, respectively. **f** SDS-PAGE analysis of the reaction mixture in the presence or absence of ACP. P and S represent the precipitate and supernatant, respectively. **g** Inhibition of FA synthesis by oleic acid. **h** Enhanced FA synthesis by the addition of liposomes. Each dot on the graphs represents individual experimental data. Error bars indicate the standard deviation of triplicate measurements. AccABCD acetyl-CoA carboxylase ABCD, ACS acetyl-CoA synthetase, ACP acyl-carrier protein, MK molecular marker.

membrane vesicles) were added during the FA synthesis (Fig. 1h). The addition of liposomes also tended to suppress the formation of aggregates even after they formed (Supplementary Fig. 5). Therefore, we envisaged that liposomes may work as a reservoir for the synthesized FAs and reduce the concentration of free or micelle FAs. A series of results obtained indicate that the synthesized FAs induce protein aggregation and disrupt their functional activities. This may be the reason why the reconstructed FA synthesis system exhibited limited productivity in the previous study[22]. Such unfavorable effect of FAs has been reported also in other proteins[24]. The addition of liposomes may

lead to the extension of the enzyme's lifetime, thus resulting in more FA synthesis.

If the same amounts of FAs were synthesized within a space of membrane vesicle (e.g., diameter 30 μm) as the in vitro reaction (300 μM), this brings that only 15% of the lipids composing the vesicle are produced (Supplementary Text 1). Even worse, not all of the FAs integrate into the vesicle membrane, because FAs have a high critical micelle concentration. Therefore, we concluded that, in order to build self-growing artificial cells, it is necessary to reconstruct the mechanism for producing more stable and inoffensive lipids and integrating them into the membrane.

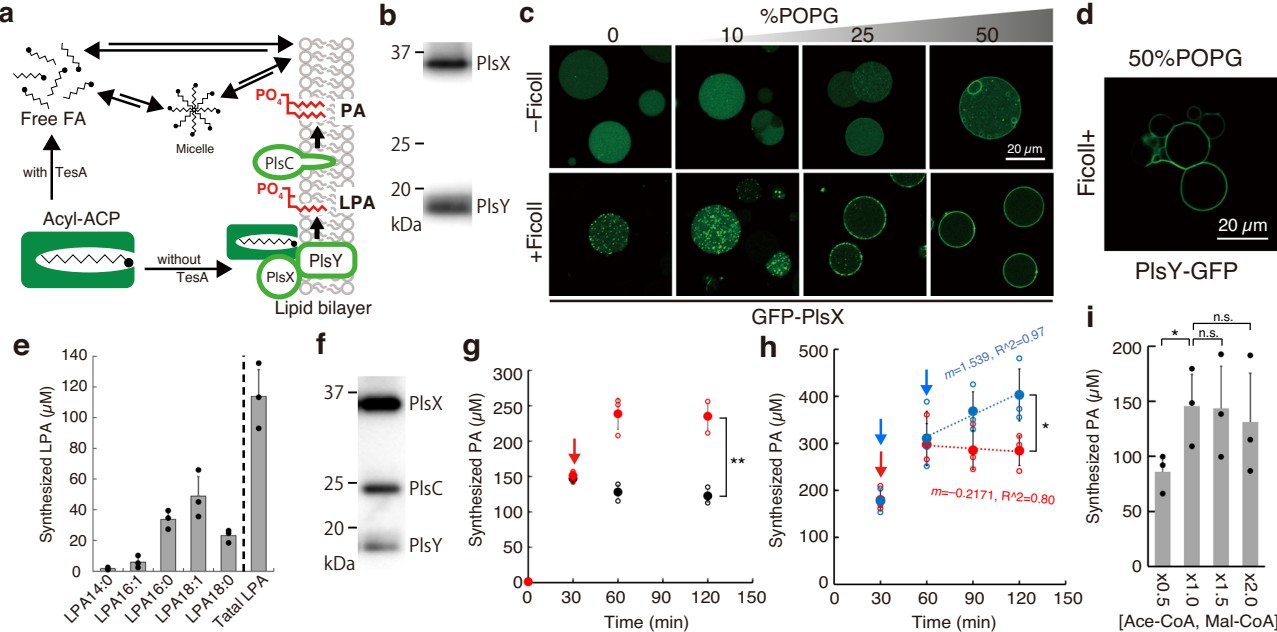

**Fig. 2 Phospholipid synthesis by cell-free synthesized acyltransferases. a** Schematic of the phospholipid synthesis pathway in a cell-free system. **b** SDS-PAGE image of cell-free synthesized PlsX and PlsY. Confocal microscopy images of **c** GFP-PlsX and **d** PlsY-GFP synthesized inside giant unilamellar vesicles (GUVs) in the presence (bottom) or absence (top) of Ficoll. The percentage of POPG (mol%) in the lipid composition of the GUV is described above the images. **e** Synthesis of lysophosphatidic acids (LPAs) by PlsX and PlsY synthesized in a bulk cell-free system. **f** SDS-PAGE image of cell-free synthesized PlsX, PlsY, and PlsC. **g** Synthesis of phosphatidic acids (PAs) by cell-free synthesized PlsX, PlsY, and PlsC. At 30 min, the resources [acetyl-CoA (Ace-CoA), malonyl-CoA (Mal-CoA), and NAD(P)H] (red) or buffer (black) were additionally supplied to the reaction mixture. **h** Increase of PA synthesis by supplying the resources at 30 and 60 min (blue) or only 30 min (red). The slopes indicate the transition in PA productions between 60 to 120 min. **i** Effect of the various concentrations of the resources supplied from the beginning of PA synthesis reaction. The concentration of 1.0× [Ace-CoA, Mal-CoA] is 2 and 4 mM, respectively. Each dot on the graphs represents individual experimental data. Double asterisk, single asterisk, and n.s. indicate that $P$ values are less than 0.01, $P$ values are less than 0.05, and $P$ values are more than 0.05, respectively. Error bars indicate the standard deviation of at least triplicate measurements.

**Design of artificial phospholipid synthesis system combining in vitro fatty acid synthesis and cell-free system.** So far, the synthesized FAs were released from ACP by TesA, thus the resulting free FAs in the aqueous reaction solution led to the inhibition of Fab enzymes. To avoid this, we designed the system to drive the reaction towards phospholipid synthesis. In bacteria, TesA does not exist in the cytosol, it locates in periplasm space. Newly synthesized FAs are directly delivered to the cell membrane in the form of FA-ACP (acyl-ACP). On the surface of the cell membrane, acyl-ACP is converted into acyl-PO4 by PlsX, then acyl-PO4 binds to a glycerol-3-phosphate by PlsY to form LPA[25,26] (Fig. 2a). If PlsX and PlsY are added to the reconstructed FA synthesis system, it is expected that the synthesized FAs are sequentially converted into LPAs, avoiding the risks of enzyme inactivation and poor membrane localization. However, these enzymes are membrane proteins that require the use of detergent during purification. Detergents melt vesicles. Therefore, we decided to generate PlsX and PlsY from their corresponding genes inside the membrane vesicles to synthesize phospholipids in cooperation with FA synthesis.

For synthesizing PlsX and PlsY, we used a reconstructed cell-free system, the PURE system[3]. Both proteins were successfully co-synthesized in a bulk (test tube) condition (Fig. 2b), resulting in about 1 μM protein production (Supplementary Fig. 6). Note that the DnaKJE chaperone was supplied to the PURE system because the solubility of PlsX is highly dependent on DnaKJE[27]. The DnaKJE is the mixture of DnaK, DnaJ, and GrpE chaperons of *E. coli*. When liposomes were supplied as localization place during the protein synthesis, 70% of the synthesized PlsX and PlsY were collected in the supernatant fraction after

centrifugation, suggesting a large part of the proteins spontaneously localized to the liposome membrane (Supplementary Fig. 7).

**Membrane localization of acyltransferases synthesized in artificial cells.** To visualize the membrane localization, we synthesized PlsX and PlsY as GFP-fusion proteins inside artificial cells. We encapsulated the PURE system with the corresponding genes inside vesicles, then incubated so prepared artificial cells. GFP-PlsX was synthesized well, however, they did not localize on the vesicle membrane (Fig. 2c and Supplementary Fig. 8). Since PlsX is a peripheral membrane protein that requires a negative charge on the membrane surface for the localization[28,29], we added 1-palmitoyl-2-oleyl-sn-glycero-3-phospho-rac-(1-glycerol) (POPG) into the lipid composition of the artificial cell membrane in the range of 0–50 mol%. We found a weak tendency of membrane localization at 25 mol% POPG, and a large part of the GFP-PlsX localized to the membrane at 50 mol% POPG (Fig. 2c and Supplementary Fig. 8). This tendency became more significant when Ficoll was together encapsulated inside the vesicles. Ficoll is a bulky polymer that is often used for inducing a molecular crowding effect inside vesicles or water-in-oil emulsions[30]. The effect of Ficoll for the membrane localization of peripheral proteins has been observed also in the case of FtsZ[6]. When Ficoll was added, fluorescence dots were observed even at 0 or 10 mol% POPG (Fig. 2c and Supplementary Fig. 8), indicating non-regulated polymerization of GFP-PlsX. Conversely, at 25 mol% POPG, most proteins localized onto the membrane. And, at 50 mol% POPG, visually all proteins localized onto the membrane. These results showed that the degree of membrane

localization of PlsX is substantially enhanced both by the negatively charged membrane surface and the molecular crowding effect. In contrast to PlsX, PlsY localized onto the membrane due to the hydrophobic interaction. We observed the efficient membrane localization of PlsY-GFP that was synthesized in the presence of Ficoll inside giant vesicles containing 50 mol% POPG (Fig. 2d and Supplementary Fig. 9).

**Lysophospholipid synthesis by coupling fatty acid synthesis and acyltransferases.** To synthesize LPA, we combined the FA synthesis system and the cell-free system synthesizing acyltransferases, PlsX and PlsY. Both acyltransferases were first synthesized in a bulk PURE system, supplemented with liposomes consisting of POPC and POPG (50:50 mol%), then mixed with Fab enzymes. The LPA synthesis was initiated by the addition of acetyl-CoA, malonyl-CoA, and NAD(P)H. We detected the synthesis of LPAs containing several types of hydrocarbon chains (Fig. 2e). The concentration of total LPAs reached 120 μM. This means that about 65% of the malonyl-CoA and 20% of the acetyl-CoA input were converted into LPA (as LPA 18:0). We also succeeded in synthesizing PlsX and PlsY in the presence of Fab enzymes, followed by the addition of the substrates to initiate LPA synthesis. The obtained data support that the cell-free synthesized acyltransferases could maintain the LPA synthesis activity on the lipid membrane in collaboration with FA synthesis. However, unfortunately, the yield of LPA (Fig. 2e) was lower than that of FA synthesis (Fig. 1c, d, g, h). We supposed that this is due to the functional regulation of PlsY by the synthesized LPA[31] to prevent the accumulation of LPA at the cell membrane. To avoid this regulation and increase productivity, we next proceeded to the synthesis of phospholipids, which are more stable and harmless lipids.

**Phospholipid synthesis by coupling fatty acid synthesis and acyltransferases.** The first phospholipid, phosphatidic acid (PA), is synthesized from LPA and acyl-ACP by PlsC (Fig. 2a), thus we additionally added the gene of PlsC in the cell-free mixture to synthesize PA. When PlsX, PlsY, and PlsC were co-synthesized (Fig. 2f), 150 μM PA was synthesized at 30 min (Fig. 2g) but the yield did not increase after that. This is supposedly due to the depletion of the resources (acetyl-CoA, malonyl-CoA, and NAD(P)H) because the amount of PA synthesized is close to the theoretical maximum that is estimated by calculation of substrate consumption. When we supplemented additional resources at the 30 min as same as the initial concentrations, the PA yield increased up to 250 μM in the next 30 min (Fig. 2g). Additionally, when the resources were again supplemented at the 60 min, the PA yield further increased to 350–400 μM at 90–120 min (Fig. 2h). Note that one PA contains two acyl-chains, thus the maximal yield of the synthesized PA is equivalent to 700–800 μM FA or LPA. This is a remarkable enhancement compared to that of FA or LPA synthesis in the previous scheme. As the PA synthesis was still increasing even after the second dose, the productivity of phospholipid synthesis in the constructed system may be governed by the concentration of acetyl-CoA and malonyl-CoA. In this experiment, 2.1 mM phospholipids, which are forming 1.3 nM liposomes, were added in the reaction mixture, thus the synthesized 400 μM PA corresponds to approximately 20% of the liposome-forming lipids. Interestingly, when we added 1.5 or 2 times more resources at the beginning of the reaction, the increase in PA synthesis was not observed and even slightly decreased (Fig. 2i). This result indicates that maintaining the optimum concentration of the substrates is important to maximize the yield and the excess substrate negatively affects lipid synthesis. We concluded that the smooth flux of sequential

reaction starting from FA synthesis to phospholipid synthesis can avoid the accumulation of synthetic intermediates and the reaction suppression by those negative feedback, thus maximizing the yield of lipid synthesis.

**Recycling of CoA towards sustainable phospholipid synthesis.** The constructed cell-free phospholipid synthesis system uses acetyl-CoA and malonyl-CoA as hydrocarbon sources. This system also produces, however, CoAs as the by-products while phospholipid synthesis: for example, 18 CoAs will be produced per one PA (18:0 or 18:1) synthesis. This is not a negligible problem when we try to synthesize sufficient phospholipids in artificial cells for reproducing the self-reproduction. A considerable amount of CoA will be accumulated. To avoid this problem and make the system more sustainable, we implemented an additional system that recycles CoA for the production of acetyl-CoA and malonyl-CoA in the constructed cell-free system. We purified acetyl-CoA synthetase (ACS) and acetyl-CoA carboxylase (Acc) ABCD (Fig. 3a). ACS synthesizes acetyl-CoA from acetic acid and CoA using ATP, and AccABCD synthesizes malonyl-CoA from acetyl-CoA and bicarbonate. The specific activities of ACS and AccDA (in the presence of 10 μM AccBC) were $425 ± 79$ and $300 ± 65$ μmol/min/mg, respectively (Supplementary Fig. 10).

When we introduced the ACS in the cell-free system and supplied CoA instead of acetyl-CoA, we detected successful synthesis of PA (Fig. 3b). This means that acetyl-CoA was synthesized from CoA by ACS and sequentially consumed as the substrate for FA synthesis. We found that the whole steps (acetyl-CoA synthesis, FA synthesis, and phospholipid synthesis) were completed within 10 min and the maximum yield was obtained when 4 mM malonyl-CoA was supplied. We also found a decrease in PA products after 10 min when 4 or 8 mM malonyl-CoA was supplied. Although the reason is unknown, this is probably due to the reverse activity of the acyltransferase. When we introduced AccABCD and supplied acetyl-CoA instead of malonyl-CoA, we also detected the PA synthesis (Fig. 3c). In this condition, a part of acetyl-CoA was converted to malonyl-CoA and consumed as the substrate for FA synthesis, together with remaining another part of acetyl-CoA. We found that the maximum yield was obtained when 8 mM acetyl-CoA was supplied. The rate of PA synthesis was not as fast as that of the reaction synthesizing acetyl-CoA, suggesting that AccABCD may be the rate-determining factor. Finally, successful PA synthesis was detected when both ACS and AccABCD were added to the cell-free system. CoA was supplied instead of acetyl-COA and malonyl-CoA (Fig. 3d). The yield and rate of PA synthesis showed a pattern similar to that of the reaction synthesizing malonyl-CoA. No PA peaks were detected by LC/MS analysis when ACS, AccDA, or AccBC was removed from the cell-free system (Supplementary Fig. 11).

We did not add acetic acid as carbon source because the cell-free system (PURE system) contains enough acetate as a counter ion of the magnesium[3]. The PURE system also contains ATP and an energy regeneration system[3], but the supplement of ATP increased the PA productivity (Supplementary Fig. 12). PA synthesis was observed even without the addition of $HCO_3^-$ (Supplementary Fig. 12). This is because $CO_2$ dissolved in the reaction solution was being converted to $HCO_3^-$ and used for malonyl-CoA synthesis. In total, the net amount of $CO_2$ ($HCO_3^-$) remains constant although it was consumed by AccC[32], because $CO_2$ was generated from the FA elongation step by FabB or FabH (Supplementary Fig. 13).

As shown above, the addition of ACS and AccABCD expanded the cell-free phospholipid synthesis system allowing for the

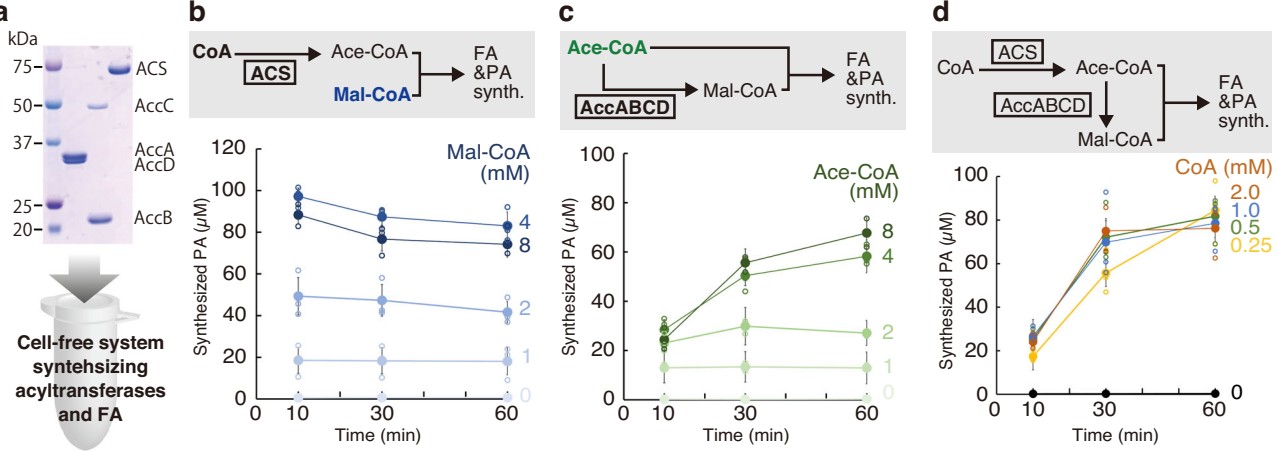

**Fig. 3 Phosphatidic acid synthesis by synthesizing acetyl-CoA or/and malonyl-CoA. a** An SDS-PAGE image of the purified acetyl-CoA carboxylase DA (AccDA), acetyl-CoA carboxylase BC (AccBC), and acetyl-CoA synthase (ACS). The positions of molecular size are described on the left side of the gel image. The purified enzymes were introduced into the cell-free system synthesizing acyltransferases and fatty acids. **b** Phosphatidic acid synthesis through the synthesis of acetyl-CoA by ACS. The reaction was initiated by the addition of CoA and various concentrations of malonyl-CoA. **c** Phosphatidic acid synthesis through the synthesis of malonyl-CoA by AccABCD. The reaction was initiated by the addition of various concentrations of acetyl-Co, which was converted to malonyl-CoA or directly consumed as the substrate of FA. **d** Phosphatidic acid synthesis through the synthesis of acetyl-CoA and malonyl-CoA by the addition of ACS and AccABCD. The reactions were initiated by the addition of various concentrations of CoA. Each dot on the graphs represents individual experimental data. Error bars indicate the standard deviation of at least triplicate measurements. FA fatty acid, PA phosphatidic acid, Ace-CoA acetyl-CoA, Mal-CoA malonyl-CoA.

production of the substrates of FA by recycling CoA. The produced substrates were consumed for the FA synthesis and, sequentially, used for the PA synthesis by the cell-free synthesized acyltransferase (Supplementary Fig. S13). Although the yield of PA synthesis (Fig. 3d) was reduced compared to the reaction using acetyl-CoA and malonyl-CoA, we decided that the constructed cell-free system enables the synthesis of phospholipids in the closed space of artificial cells.

**Design and construction of artificial cells synthesizing phospholipids.** To perform phospholipid synthesis inside artificial cells, we designed the internal cell-free mixture that composes of four subsystems: CoA recycling system, FA synthesis, transcription-translation, and phospholipid synthesis (Fig. 4a). In this system, Fab enzymes, ACP, ACS, and ACCs are encapsulated as soluble recombinant proteins that exhibit the activities within the vesicle lumen. On the other hand, PlsX, PlsY, and PlsC are synthesized from the genes within the vesicle lumen because these are membrane-localized proteins. These successive reactions start with the production of acetyl-CoA (ace-CoA) and malonyl-CoA (mal-CoA), allowing the synthesis of FA. The FA synthesis is driven by the electron transfer of NAD(P)H supplied from outside of the vesicles. The synthesized FA as a binding state with ACP is converted into lysophosphatidic acid (LPA) by PlsX and PlsY, and sequentially converted into PA by PlsC. Therefore, the synthesized phospholipids are directly integrated into the membrane of the artificial cell. The presence of PA on the membrane can be detected by the binding of a fluorescent probe protein, GFP-Spo (Fig. 4a). The GFP-Spo is a fusion protein of GFP and Spo20, which contains an amphipathic motif that specifically recognizes PA on the membrane environment[33].

Before constructing the designed artificial cells, we first verified whether CoA and NAD(P)H inhibit protein synthesis because both factors have to be encapsulated together with all components of the PURE system within a vesicle. We found that the addition of 4 mM NAD(P)H in the PURE system reduced the synthesis of the three acyltransferase proteins to less than 50% (Supplementary Fig. 14). On the other hand, the addition of CoA

did not reduce the synthesis, or even slightly enhanced it for unknown reasons (Supplementary Fig. 15).

Next, we examined the membrane permeability of NADPH and CoA. An NADPH-sensitive fluorescent probe was encapsulated in giant vesicles composed of 50 mol% POPG and 50 mol% POPC, then 4 mM NADPH was added to the exterior. After 90 min of incubation, slight fluorescent was detected from the lumen of the vesicles (Supplementary Fig. 16). This was almost the same as the negative control that supplied H₂O instead of NADPH, indicating that NADPH did not translocate the membrane. However, when the POPG ratio of the vesicle membrane was reduced to 20 mol%, we detected a significant fluorescence response (Supplementary Fig. 16). The fluorescence level increased for at least 90 min. These results indicate that NADPH is able to permeate the vesicle membrane if the negative charge on the membrane surface is moderate, i.e., 20 mol% POPG. The same permeability test was performed for CoA by using a CoA-sensitive fluorescence probe. Unlike NADPH, when 1 mM CoA was added to the outside, no fluorescence was observed from the inside of the vesicles in neither case 50 mol% nor 20 mol% POPG, indicating that CoA cannot penetrate the vesicle membrane (Supplementary Fig. 17).

Based on the obtained results of the protein synthesis inhibition and the membrane permeability, we employed a strategy that first synthesizes the acyltransferase proteins in the presence of CoA but in the absence of NADPH, then, after the protein synthesis, we supply NADPH to initiate the internal FA synthesis. The reaction mixture consists of the PURE system and the enzymes for FA synthesis (Supplementary Table 2) were encapsulated inside giant vesicles composed of 80 mol% POPC and 20 mol% POPG. The prepared artificial cells were once washed with the buffer solution to remove the leaked reaction components from the droplets that failed the membrane vesicle formation. Furthermore, RNaseA was supplied to inhibit a possible synthesis of acyltransferases outside, which is caused by the factors leaked from burst artificial cells during the incubation. Phospholipid synthesis was initiated by the addition of NADPH, then GFP-Spo was mixed to identify the PA on the membrane of reacted artificial cells.

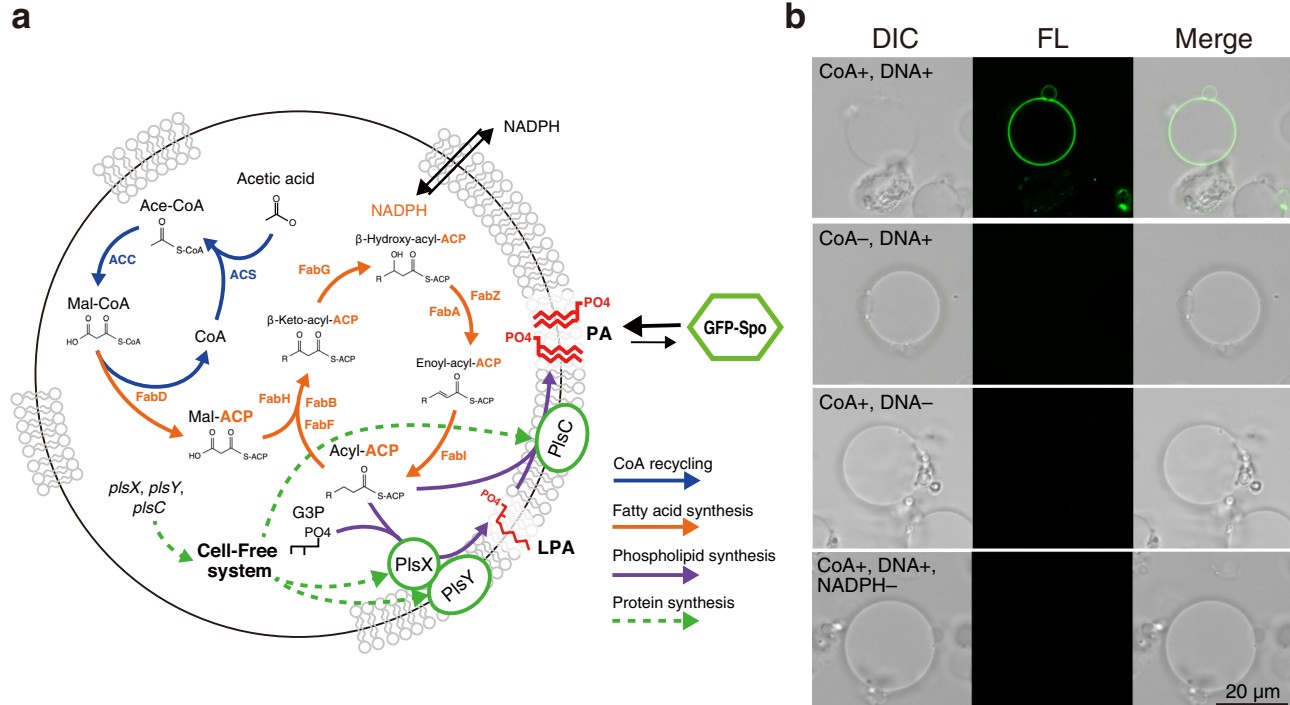

**Fig. 4 Phospholipid synthesis inside artificial cells. a** Schematic of the phospholipid-synthesizing artificial cell. GFP-Spo was added to the exterior of the cells to visualize the synthesized PA at the membrane. **b** Confocal microscopy images of the artificial cells treated with GFP-Spo after the lipid synthesis reaction. The artificial cells missing CoA, DNA, or NADPH were also tested for the controls. Wide-filed images of the artificial cells are shown in (Figs. S18). CoA coenzyme A, Ace-CoA acetyl-CoA, Mal-CoA malonyl-CoA, Mal-ACP malonyl-ACP, ACP acyl-carrier protein, G3P glycerol-3-phosphate, LPA lysophosphatidic acid, PA phosphatidic acid.

The reacted artificial cells exhibited a significant fluorescence of GFP-Spo on the vesicle membrane (Fig. 4b and Supplementary Fig. 18). When CoA was removed from the system, resulting in a lack of acetyl-CoA synthesis, no lipid synthesis occurred thus no fluorescence was observed (Fig. 4b). The synthesis of PA was further confirmed by omitting the template DNAs of acyltransferases or the addition of NADPH. The result showed again no fluorescence of GFP-Spo on the membrane (Fig. 4b). We also confirmed the successful PA-synthesizing artificial cells in small magnification images with many vesicles (Supplementary Fig. 18). By counting the labeled vesicles, we estimated that approximately 10–20% of the artificial cells show significant fluoresce that was assumed to have completely performed the reaction from the acyltransferase syntheses to PA synthesis. The GFP-Spo used in this study showed a significant binding onto the membrane when PA was present at 10 mol% (Supplementary Fig. 19). If the encapsulated cell-free system produced 100 mM PA as same as the in vitro reaction (Fig. 3d), it corresponds to 10 mol% of the phospholipids forming the artificial cell membrane (in the case of 30 μM diameter) (Supplementary Table 1). Therefore, the results obtained by the microscopy observation are consistent with the theoretical estimation based on the results of in vitro experiments, supporting the successful phospholipid synthesis in artificial cells.

## Discussion
The construction of artificial cells based on cell-free systems may lead to the development of new biological tools that will be a platform for the study of cellular biological processes or a fundamental technology for biomedicine[34,35]. One of the bottlenecks that impede the development of artificial cells is the delay in reproducing the self-reproduction ability of cells. In particular, the difficulty of building autonomous membrane growth is the core issue[8]. Therefore, it is important to achieve the synthesis of

phospholipids in the artificial cell and, for that purpose, the construction of a cell-free system that promises high productivity in phospholipid synthesis is crucial.

In the previous studies with biochemical approaches, phospholipids synthesis in vitro or in artificial cells has been demonstrated by expressing the enzymes involved in phospholipids synthesis. However, the yield of synthesized phospholipids in such manners was around 30 μM[18]. This corresponds to only a few percent of the lipids of the mother cell membrane and, therefore, it was far from the concentration required for the self-reproduction of a cell. The reason for the low productivity is thought to be supplying acyl-CoA as a source of hydrocarbon chains. In this scheme, a highly concentrated acyl-CoA has to be encapsulated for achieving enough lipid synthesis. However, since the critical micelle concentration of acyl-CoA (e.g., palmitoyl-CoA) in physiological conditions is tens μM[36], most of the acyl-CoAs form semi-stable micelle structures in the order of mM concentration, thus the rate consumed as the substrate of acyl-transferase would be very slow. Furthermore, with such high concentrations of acyl-CoA, there are concerns about the inhibition of protein synthesis and/or deformation of membrane vesicles.

In our system, we were able to avoid this problem by employing the synthesis of hydrocarbon chains (FAs), resulting in an order of magnitude higher synthesis of phospholipids. The lipid synthesis pathway is known as a highly regulated process. Indeed, we have witnessed the surfactant-like property of FAs (Fig. 1g), a possible inactivation of enzymes by LPAs (Fig. 2e), and inhibition of lipid synthesis by excessive amounts of acetyl-CoA and malonyl-CoA (Fig. 2i). We could overcome these unfavorable regulations by smoothly completing lipid synthesis from the harmless small molecular compound to the stable lipid molecules. Nevertheless, the productivity of phospholipid synthesis decreased when we

synthesized phospholipids from CoA by adding ACS and AccABCD. This implies that there may be another regulation in the steps from CoA to malonyl-CoA, perhaps by an acyl-ACP intermediate. By solving this regulation, we would achieve a sufficient yield of phospholipid synthesis for self-reproduction.

In addition to PlsXYC, by synthesizing further downstream enzymes, we can produce various phospholipids beyond PA, including phosphatidylcholine and phosphatidylglycerol, which form the membrane of the mother artificial cells. One of the advantages of our system is that the system is based on the gene expression system. Therefore, theoretically, phospholipid synthesis can be sustained even in the divided cells by expressing the corresponding enzymes. Another advantage is we can produce phospholipids avoiding the accumulation of the by-product by reusing the used CoA and by recruiting acetic acid as a carbon source. Conversely, the limitation of our system is the lack of machinery for importing the substrates (e.g. glycerol-3-phosphate) from the outside of the vesicles. The membrane leakage of some molecules from inside to outside is also problem. This might be solved by changing the lipid composition of the membrane. In the present experiments, the constructed artificial cells consumed pre-encapsulated substrates and membrane-permeated cofactors by natural diffusion. We found that NADPH permeates the vesicle membrane consisting of 20 mol% POPG. Although the detailed mechanism is unknown, maybe NADPH interacts with the membrane and forms transient pores by defecting the stability of the membrane. Considering the construction of a sustainable artificial cell system, it is necessary to implement transporters for those substrates and cofactors on the artificial cell membrane. The reduction of $NAD(P)^+$ should also be considered. This could be solved by combining our system with glycolysis to reduce $NAD(P)^+$ and produce glycerol-3-phosphate.

Another point, it has been known that there is a certain amount of residual oil between the leaflets of giant vesicle membrane in droplet transfer method. Although we cannot eliminate the possibility of the residual oil in our artificial cell membrane, we think that the effect of the oil on the activities of the membrane-associated proteins is not significant, because the oil localizes at the north pole of the vesicle rather than equally distributes within the lipid bilayer. This is because the oil is lighter than water. Contrary, membrane-associated proteins are dynamic on a lipid membrane due to the fluid mosaic model. Therefore, we think that the effect of the residual oil on membrane-associated proteins may be limited.

The population of PA-synthesizing artificial cells detected by the membrane binding of GFP-Spo was not high (Fig. 4b and Supplementary Fig. 18). About this, there is a concern about whether the PAs, synthesized inside and localized onto the inner leaflet of the vesicle membrane, did flip-flop to the outer leaflet that is accessible by the GFP-Spo. More detailed analysis of the physicochemical property of the membrane will be needed to clarify how the flip-flops of lipid in the growing membrane affect the shape change of vesicles.

In the present study, we could not observe morphological changes of the artificial cells. This may be due to the low production of phospholipids (max. 100 μM) originating from the acetyl-CoA synthesis. The cascading reaction from FA synthesis to PA synthesis seems no problem as far as the adequate amounts of substrates are supplied. However, there is a rate-limiting step in the malonyl-CoA synthesis, which is probably regulated by the acyl-ACP intermediates[37]. In fact, the malonyl-CoA synthesis rate of the acetyl-CoA carboxylase reached to equilibrium state before depleting the substrate, in contrast to the acetyl-CoA synthesis rate of the ACS that lasted to the maximum yield (Supplementary Fig. 10). To increase the productivity of lipid synthesis, it may be necessary to replace the acetyl-CoA

carboxylase with an enzyme derived from another species that can circumvent the regulation by acyl-ACP intermediates. It should be noted that the shape of the membrane vesicle easily changes, regardless of the internal biochemical reactions, due to a decrease in internal volume by a change in osmotic pressure. To avoid false-positive results and to make accurate time-lapse observations, defined experimental settings using microfluidics devices, flow cytometry, and super-resolution microscopy are important. By overcoming these challenges, it will be possible to construct artificial cells that can autonomously grow and divide like living cells. Our research would be a basis toward that end.

## Materials and Methods

**Materials**. 1-palmitoyl-2-oleoyl-glycero-3-phosphatidylcholine (POPC), 1-palmitoyl-2-oleoyl-sn-glycero-3-phospho-(1′-rac-glycerol) (POPG), 1-myristoyl-2-hydroxy-sn-glycero-3-phosphate (14:0 Lyso PA), 1-palmitoyl-2-hydroxy-sn-glycero-3-phosphate (16:0 Lyso PA), 1-stearoyl-2-hydroxy-sn-glycero-3-phosphate (18:0 Lyso PA), 1-oleoyl-2-hydroxy-sn-glycero-3-phosphate (18:1 Lyso PA), 1,2-dipalmitoyl-sn-glycero-3-phosphate (DPPA), 1-palmitoyl-2-oleoyl-sn-glycero-3-phosphate (POPA), and 1,2-dioleoyl-sn-glycero-3-phosphate (DOPA) were purchased from Avanti Polar lipids. Fatty acid LC/MS mixture (#17942) was purchased from Funakoshi (Japan). $^{13}$C-Acetyl-CoA, $^{13}$C-Malonyl-CoA, Acetyl-CoA, Malonyl-CoA, NADH, NADPH, and sn-glycerol-3-phosphate were purchased from Sigma Aldrich. PURE*frex* 2.0 and its Sol.I buffer, which was customized for the volume down, and DnaKJE mix were given or purchased from GeneFrontier (Japan).

**Overexpression and protein purification**. All fatty acid binding (Fab) enzymes, acyl carrier protein (ACP) and thioesterase I (TesA) were overexpressed (Supplementary Table 3) and purified as previously reported by Yu et al.[22] with some modifications. Briefly, for FabA/B/D/F/G/H/I/TesA, the transformed cells were cultivated in 1 L LB medium supplied 50 μg/mL kanamycin at 37 °C. At $OD_{600}$ 0.6–0.8, 0.1 mM Isopropyl-β-D-thiogalactoside (IPTG) was introduced for induction, and the cultivations were continued for 3 h. The cells were then collected by centrifugation 5500 × g for 20 min at 4 °C, washed with Buffer A (50 mM Tris-HCl (pH 7.6), 2 mM dithiothreitol (DTT), 10% glycerol), and stored at −80 °C until use. The frozen cells were dissolved in 35 mL Buffer A on ice and disrupted by sonication (5 s on, 3 s off power 40%, total 5 min). The unbroken cells and debris were removed by centrifuging 30 k × g for 1 h at 4 °C. The resulting supernatants were collected and injected into AKTA™ pure equipped HisTrap HP column (GE Healthcare), which was pre-equilibrated with Buffer A. The 6HisTagged proteins were eluted by a gradient of Buffer B (Buffer A with 1 M Imidazole) from 3% to 50%. The target proteins were collected and desalted using Amicon (with their corresponding cutoff size) (Millipore) before subjecting to the HiTrap Q column (GE Healthcare). The injected proteins were washed and eluted by a gradient of Buffer C (Buffer A with 1 M NaCl) from 0% to 100%. Finally, the proteins were concentrated by Amicon and their concentration was determined with Bradford (Takara Bio) or BCA (Thermo Fisher Scientific) protein assay kit. In general, the purified proteins are stable for at least a few months at −80 °C.

ACP was co-expressed with a 4′-phosphopantetheinyl transferase (SFP) because ACP needs post-translational modification to form Holo-type ACP (Holo-ACP). The source of the *sfp* gene is *Bacillus subtilis*, and the codon was optimized for the *E. coli* expression system when we ordered it as a synthetic gene (Supplementary Table 4). The gene was inserted between NcoI and BamHI sites of the pCDF-1b (Novagen) vector using the primers sfp-fw and sfp-rv (Supplementary Table 5), resulting in pCDF-sfp (Supplementary Table 3). The *E. coli* BL21 (DE3) cells harboring pTL14, which contains the *acp* gene, and pCDF-sfp were cultivated in TB (Terrific Broth) medium supplied kanamycin and streptomycin until $OD_{600}$ arrived at 0.6. After the induction with 1 mM IPTG, cells were collected at $OD_{600}$ 1.9 and washed with 50 mM HEPES-KOH (pH7.6). The obtained cell pellets were dissolved in Buffer A and disrupted by French Press at 500 bar (7000 psi) three times. After removing cells unbroken and debris by centrifugation, the supernatant was subjected HisTrap HP column as in the case of the Fab proteins, using the Buffer A and B containing 300 mM NaCl. The target proteins were collected and desalted before subjecting MonoQ 4.6/100PE column (GE Healthcare) using Buffer A and C as shown in Fab proteins. The collected target proteins were desalted by nap-5 and concentrated by Amicon. Concentrations of the protein were determined by Protein Assay (BioRad). MALDI-TOF-MS was used to confirm the Holo-type of the purified ACP.

Because FabZ tends to aggregate, we modified the expression and purification method as follows. The expression plasmid pXY-FabZ was introduced to *E. coli* C43 (DE3) strain cells. The cells were cultivated at 37 °C in 1 L LB medium supplied kanamycin until the $OD_{600}$ 0.6, then, 1 mM IPTG was added followed further 2 h of cultivation. The resulting cells were collected and washed with 50 mM HEPES-KOH (pH 7.6), then stored at −80 °C until use. The cell pellets were resuspended in Buffer D (100 mM potassium phosphate (pH 7.4), 300 mM NaCl, 1 mM DTT) and disrupted by French Press as same as ACP. After removing

cells unbroken and debris, the supernatants were subjected HisTrap HP column with Buffer D and E (Buffer D with 1 M Imidazole). The FabZ was eluted with a gradient of Buffer E from 10% to 100%. The fractions containing FabZ were collected and desalted by repeating dilution with Buffer D and concentrating by Amicon. Note that the purified FabZ should not be concentrated to a high concentration to prevent aggregation. We did not proceed to further purification because FabZ is unstable, and only HisTrap purification is enough for purity. When determination of the FabZ concentration by the Protein Assay is difficult, the concentration was defined by comparing the band intensity of FabZ on SDS-PAGE with the standard protein which concentration had already been determined. Interestingly, the aggregated FabZ was dissolved by mixing with ACP. This might be due to the complex formation of FabZ and ACP that has been reported by Zhang et al.[38]. Therefore, FabZ should be stored as a mixture with ACP in the molar ratio of 1:30 (FabZ: ACP).

Acetyl-CoA synthetase (ACS) was overexpressed from the synthetic gene (Supplementary Table 4) located between NdeI and XhoI sites of the pET-28a vector, resulted in pET28a-acs, introduced in E. coli C43 (DE3) cells, referring to a previous report[39]. Cells were cultivated at 37 °C in LB medium supplied kanamycin until the $A_{600}$ 0.4, then, 0.4 mM IPTG was added followed by additional three hours of cultivation. The resulting cells were collected and washed with 100 mM potassium phosphate (pH 7.4), then stored at −80 °C until use. The frozen cell pellet was dissolved in Buffer D and disrupted by French Press as in the case of ACP. After removing cells unbroken and debris, the supernatants were subjected by AKTA Avant 25 (GE Healthcare) equipped HisTrap HP column with Buffer D and E. The ACS was eluted by the gradient of Buffer E from 10% to 50% for 20 min by a flow rate 2.5 ml/min. The fractions containing ACS were collected and desalted by repeating dilution with Buffer E and concentrating by Amicon 30 kDa cut off. The resulting sample was further purified by MonoQ 4.6/100PE column (GE Healthcare) using Buffer F (50 mM Tris-HCl (pH 8.0), 10% glycerol) and Buffer G (Buffer F with 1 M NaCl). ACS was eluted by the gradient of Buffer G from 0% to 100% for 20 min by a flow rate 1.0 ml/min. The collected ACS was desalted and concentrated using Amicon. We did not cut off HisTag because the protein aggregates.

AccBC complex was purified as previous reports[32], with slight modifications. The polycistronic genes for AccB and AccC were obtained as a synthetic gene (Supplementary Table 4) with codon optimization for the E. coli expression system (FASMAC), which inserted between NdeI and XhoI sites of the pET28a vector, resulted in pET28a-accBC. In this construct, a 6HisTag was inserted at the N-terminus of the accB gene. The gene of birA, biotin-[acetyl-CoA-carboxylase] ligase, is also codon-optimized and inserted in the pCDF-1b vector using NcoI and XhoI sites, resulted in pCDF-birA. Both pET28a-accBC and pCDF-birA were co-introduced into C43 (DE3) E. coli cells (Lucigen) and then cultivated at 37 °C in LB medium supplemented with kanamycin and streptomycin. At the OD$_{660}$ 0.4, 0.5 mM IPTG and 2 μM biotin were added, then, followed by further cultivation for four hours. Cells were collected and dissolved in Buffer H (20 mM potassium phosphate (pH 8.0), 500 mM NaCl, 2-mercaptoethanol, protease inhibitor cocktail), then disrupted by French Press as described above. After removing cells unbroken and debris, the lysate was applied to the HisTrap HP column with Buffer H and I (Buffer H with 1 M imidazole). AccBC protein was eluted as complex by the gradient of Buffer I from 2% to 100% for 20 min by 5.0 ml/min flow rate. The fractions of AccBC were collected and desalted using Amicon (10 kDa cut off) and Buffer J (50 mM Tris-HCl (pH 8.0), 100 mM NaCl, 10% glycerol). The desalted proteins were further purified by the MonoQ column with Buffer J and K (Buffer J with 1 M NaCl). AccBC was eluted by the gradient of Buffer K from 0% to 100% for 20 min by 1.5 ml/min flow rate. The collected AccBC fractions were buffer exchanged for Buffer L (20 mM Tris-HCl (8.0), 300 mM NaCl, 10% glycerol) while being concentrated by Amicon (10 KDa cut off). The concentration was determined by the BCA protein assay kit. Finally, the efficiency of biotinylation of the purified AccB biotin carboxyl carrier protein of acetyl-CoA carboxylase was confirmed. One microgram of AccBC sample was dissolved in 25.5 μL 50 mM ammonium bicarbonate (pH 8.0), then 1.5 μL fresh 500 mM DTT was added. After boiling at 95 °C for five minutes, 3 μL 500 mM iodoacetamide was added, followed by a 20 min incubation at room temperature. For digestion of the protein, 0.5 μL 5 ng/μL Gluc-C (Promega) was added and incubated for three hours at 37 °C, followed by additional supplying 0.5 uL Gluc-C and incubation at 30 °C overnight. The digestion reaction was terminated by the addition of 3 μL of 50% trifluoroacetic acid (TFA). The sample was then purified by a C18 spin column (Thermo Fisher) following the manufacturer's protocol. The eluted sample was dried up and dissolved in 5% acetonitrile solution supplied 0.5% TFA, before analyzing by NanoLC-MS (Thermo Fisher). From the obtained data, the area intensity of the biotinylated fragment (AMkMMNQIE, $m/z$ = 661.29232, $z$ = 2, where k = biotinylated lysine residue) was compared with that of a non-biotinylated fragment (AMKMMNQIE, $m/z$ = 548.25360, $z$ = 2). Generally, more than 50% of the purified AccB is biotinylated.

AccDA was expressed from pACS275[32] introduced in C43 (DE3) E. coli cells. pACS275 contains the genes of accD, where a 6HisTag sequence was added at the C-terminus, and accA. The protein purification was carried out following a previous report[32], with slight modifications. The cells were cultivated at 37 °C until the OD$_{660}$ reached 0.5, then 0.5 mM IPTG was added, then cells were further cultivated at 25 °C for 4 h. Cells were collected and washed with Buffer M (20 mM Tris-HCl (pH 8.0), 500 mM NaCl, 10% glycerol). For disruption of the cells, 2 μL 250 U/μL Benzonase (Merck) and 8 mg lysozyme were added into a 40 mL cell

solution and kept on ice for one hour. After three times freeze-and-thaw, unbroken cells and debris were removed by centrifugation and the supernatant was passed a 0.45 μm filter. The resulting lysate was supplemented 25 mM (final concentration) imidazole, then subjected by the HisTrap HP column with Buffer M and N (Buffer M with 1 M imidazole). The sample was washed with 5% Buffer N for 10 column volume by flow rate 5.0 mL/min, then eluted by the gradient of Buffer N from 5% to 25% for 20 column volume. AccDA fractions were collected and diluted five times with Buffer O (20 mM Tris-HCl (pH 8.0) and 10% glycerol). The resulting sample was loaded to the HiTrap Capto Q column (Cytiva) and washed 10% Buffer P (Buffer O with 1 M NaCl) for 10 column volume. AccDA sample was eluted by the gradient of Buffer P from 10% to 100% for 20 column volume. The collected fractions were concentrated and buffer exchanged with Buffer L using Amicon (10 kDa cut off).

The compositions of all buffers used for protein purifications are shown in Supplementary Table 6.

**Liposome preparation.** Small size liposomes were used for in vitro phospholipid synthesis reaction. 40 mg POPC powder or a mixture of 20 mg POPC and 20 mg POPG powder was dissolved in chloroform within a round-bottom 50 mL flask and briefly processed by bath sonication. Then chloroform was removed by rotary evaporator and placed under low pressure within a desiccator for overnight to remove the residual chloroform completely. The resulting lipid film was dissolved with 1 mL 50 mM HEPES-KOH (pH 7.6) by vortexing and sonicating. The lipid solution was next processed by micro extruder using 1 μm and 200 nm pore size polycarbonate membrane in a stepwise fashion. The prepared liposome solution was divided into micro tubes and stored at −80 °C. The liposome sample is generally stable for at least years, but it should be treated by a bath sonication before use.

**In vitro fatty acid synthesis.** In vitro fatty acid synthesis reaction was prepared as described in Supplementary Table 7, based on the report by Yu et al.[22]. 13C-stable isotope-labeled acetyl-CoA and malonyl-CoA were used as substrates to distinguish the products from environmental fatty acids. The reaction was generally carried out at 30 °C for 30 min. The resulting reaction mixtures were diluted to 20-fold with 50% MeOH and centrifuged at $20,000 \times g$ for 30 min at 16 °C. The resulting supernatants were collected and placed in glass vials. Synthesized fatty acids were analyzed by Shimadzu LCMS-2020 system equipped with InertSustain® Phenylhexyl column (3 μm, 2.1 mm Cat. No. 5020-89128, GL Science, Japan) with a guard column (GL Science, Japan). Fatty acids were separated in 60% B isocratic elution mode using mobile phase A (20 mM ammonium acetate) and B (100% acetonitrile). The column oven was 35 °C, and the flow rate was 0.14 mL/min. Elution was analyzed by selected ion monitoring (SIM) mode monitoring $m/z$ of 211.9 ($C^{13}$12:0), 241.2 ($C^{13}$14:0), 269.2 ($C^{13}$16:1), 271.2 ($C^{13}$16:0), 299.2 ($C^{13}$18:1), and 301.3 ($C^{13}$18:0) (Supplementary Table 8) in the negative-ion mode. For the quantification, the area counts obtained from the peaks were converted into fatty acid amounts using the standard curve made from the analysis of the commercially available standard fatty acid mixture with defined concentrations. The standard fatty acids samples were measured each time of the experiment. An example of the standard curve is shown in Supplementary Fig. 20.

**Preparation of template DNA for cell-free protein synthesis.** The genes expressed in the cell-free system (the PURE system) were summarized in Supplementary Table 9. All genes were prepared as plasmid DNA by In-Fusion cloning or linear DNA by PCR (Supplementary Table 10) using the designed primer sets (Supplementary Table 5) and template DNAs (Supplementary Table 4). The E. coli gene of plsX and plsY was obtained from K12 strain cells and inserted into the pET28a vector. Using the resulting plasmids, we introduced a gene for super-folder green fluorescence protein (sfgfp)[7] to the upstream of plsX or the downstream of plsY by In-Fusion to generate a GFP-PlsX$_{wt}$ or PlsY$_{wt}$-GFP plasmid, respectively (see Supplementary Table 9 #7&8). These were used for expression in GUV.

Because the productivity of PlsX in the PURE system was not so high, we reduced the GC-contents just after the initial codon by introducing silent mutations by PCR (see Supplementary Table 9 #1). We used this liner DNA for cell-free LPA or PA synthesis. In the same manner, we also prepared six histidine-tagged plsX for the analysis of western-blotting, using an anti-HisTag antibody (see Supplementary Table 9 #4). The E. coli gene of plsY for cell-free LPA or PA synthesis was obtained as a synthetic gene with codon optimization for the E. coli expression system (see Supplementary Table 9 #2). As same as plsX, a six histidine-tagged plsY was prepared by PCR (see Supplementary Table 9 #5). The E. coli gene of plsC was obtained as a previously prepared plasmid pBT302_plsC[19] (see Supplementary Table 9 #3). This was used for cell-free PA synthesis and for preparing a six histidine-tagged plsC DNA (see Supplementary Table 9 #6).

**Cell-Free Protein synthesis.** Cell-Free gene expressions were performed using PUREfrex 2.0 (GeneFrontier, Japan) following the manufacturer's protocol. In general, linear template DNA(s) of plsX, plsY, and plsC were introduced into the reaction mixture supplied liposomes, sucrose, and DnaKJE chaperone mixture (DnaKJE mix)[27] which highly affects to the solubility of PlsX[27] (eSol: http://www.tanpaku.org/tp-esol/index.php?lang=en) (Supplementary Table 2). For the gene

expression to LPA or PA synthesis, the custom Sol. I, which was adjusted to reduce the volume from 10 uL to 6 μL by GeneFrontier company, was used to make a space to add the additional components, such as glycerol-3-phosphate (G3P) and fatty acid enzymes mixture (FA mixture). The FA mixture was prepared as shown in Supplementary Table 2, of which 2 μL was used to make 20 μL of the reaction mixture. For the gene expression to synthesize PA in the CoA recycling system, ACS, AccBC, and AccDA were additionally introduced using the water space of the mixture. In all cases, gene expression reactions were carried out at 37 ˚C for 2 h using a thermal cycler.

**Solubility assay**. The ratio of membrane localization of the synthesized PlsX and PlsY were assessed as in a previous report[40], using the *plsX*-6His and *plsY*-6His genes. Proteins were synthesized in the presence or absence of liposomes consisting of 50% POPC and 50% POPG. The proteins in the supernatant or precipitation fractions were analyzed by western blotting using an anti-HisTag monoclonal antibody-conjugating horseradish peroxidase.

**In vitro phospholipid synthesis**. After the gene expression, the reaction mixture was supplemented with NADH, NADPH, acetyl-CoA, malonyl-CoA, and potassium phosphate buffer as described in Supplementary Table 2. For LPA or PA quantification by LC/MS, we used [13C]-labeled Acetyl-CoA and Malonyl-CoA. LPA or PA synthesis was performed at 30 ˚C. The resulting mixture was pretreated as same as a previous report[18] before analyzing by LC/MS. Briefly, 5 μL of sample synthesizing LPA or PA was diluted with 45 μL or 495 μL 100% MeOH solution including 2 mM acetylacetone, respectively. The diluted sample was sonicated for 10 min and then centrifuged at $16,000 \times g$ for 5 min at 15 ˚C. The supernatant was passed a 0.2 μm filter (Millex®-LG) and injected into LCMS-2020 (Shimadzu). An internal standard was added to assess lipid extraction efficiency, which contained 0.025 ng/μL each of DPPA, POPA, and DOPA in MeOH.

**LC/MS analysis for synthesized phospholipids**. For separation of synthesized LPAs or PAs, a metal-free column L-column2 or L-column3 (CERI, JAPAN) was used to avoid adsorption of phospholipids on the metal surface of the column. The LC/MS analysis method is based on Blanken et al.[18] with slight modifications. Mobile phase A (water with 0.05% ammonium hydroxide and 2 mM acetylacetone) and B (80% 2-propanol, 20% acetonitrile, 0.05% ammonium hydroxide, and 2 mM acetylacetone) were used for the isolation of LPA or PA. For LPA, column oven at 55 ˚C, flow rate 0.2 mL/min, and %B 30 were applied. 5 μL samples were injected, then eluted by a gradient from 30% to 100%B in 30 min followed. Elution was directly sprayed into the MS to analyze the products in negative mode. The *m/z* values of 395.0 (LPA13C14:0), 393.2 (LPA13C14:1), 425.1 (LPA13C16:0), 423.1 (LPA13C16:1), 455.1 (LPA13C18:0), and 453.1 (LPA13C18:1) were monitored as 13C-labeled products in SIM mode as shown in Supplementary Table 8. For PAs, when L-colum2 was used, the column oven at 60 ˚C, the flow rate 0.2 mL/min, and %B 50 was applied. When L-column3 was used, the column oven at 45 ˚C, the flow rate 0.15 mL/min, and %B 50 was applied. In both cases, 5 μL samples were injected, then eluted by a gradient from 50 %B to 100 %B in 30 min followed. The *m/z* values of 679.5 (PA13C16:0/16:0), 677.5 (LPA13C16:0/16:1), 709.5 (LPA13C16:0/18:0), 707.5 (LPA13C16:0/18:1), 675.5 (LPA13C16:1/16:1), 705.5 (LPA13C16:1/18:1), 739.5 (LPA13C18:0/18:0), 737.5 (LPA13C18:0/18:1), and 735.5 (LPA13C18:1/18:1) were monitored as 13C-labeled products in negative SIM mode as shown in Supplementary Table 8.

**Formation of giant vesicles by droplet transfer method**. The formation of giant vesicles encapsulating the cell-free system was performed as described in Shimane et al.[41]. Briefly, a 50 μL of lipid mixture dissolved in chloroform was transferred into a grass tube and dried up by blowing nitrogen gas during vortexing to obtain lipid film. For example, to make 80 mol% POPC and 20 mol% POPG vesicles, a lipid mixture containing 40 mM POPC and 10 mM POPG was used for the lipid film formation. The dried lipid film was dissolved in 500 μL mineral oil by processing heating at 70 ˚C for 1 min, followed by vigorous vortexing for 30 s. The lipid-oil mixture was heated again for 1 min and vortexed for 3 min. So prepared lipid-oil mixture was used for making water-in-oil droplets by mixing with 20 uL of inner solution and vigorous vortexing for 30 min. The prepared droplets solution was transferred over a 300 uL of the outer solution (the same composition as 1× PURE*frex2.0* buffer lacking tRNA mix, creatine phosphate, formyl doner, and NTPs; provided by GeneFrontier company) containing 300 mM glucose in a 1.5 mL microtube and, then, centrifuged at 10,000 x *g* for 5 min at room temperature. After removing the upper oil phase by pipetting, the fraction of the giant vesicles precipitated at the bottom of the tube was carefully collected for 20 μL. To wash away the inner solution that leaked out from the droplets failing to form vesicles, the collected vesicle fraction was mixed with 180 μL of outer solution and precipitated again. The quality and population of the formed giant vesicles were checked by inverted microscopy equipped with a phase-contrast unit (Nikon IX73).

**Phospholipid synthesis inside giant vesicles**. The internal reaction mixture was prepared as described in Supplementary Table 2 (see For PA synthesis inside GUVs), then mixed with 12% (w/v) Ficoll PM70. This was encapsulated inside

giant vesicles as described above with the outer solution. The collected 20 uL vesicle solution (i.e., artificial cells) was mixed with 6 μL of a feeding solution; containing 1×PURE*frex2.0* buffer, 13 mM ATP, 6 mM glycerol-3-phosphate, 4.8 mM CoA, 42 ng/μL RNaseA, and 300 mM glucose. Protein synthesis was performed at 37 ˚C for 3 h. After the protein synthesis within the artificial cells, the internal phospholipid synthesis was initiated by the addition of 5 μL of 48 mM NADPH dissolved in the outer solution. Phospholipid synthesis was carried out at 37 ˚C for 1–9 h. The resulting 30 μL artificial cell sample (1 μL was used for microscopy check) was mixed with 5 μL of 170 ng/μL GFP-Spo protein to stain the vesicle membrane and, then, incubated at 37 ˚C for 30 min followed by an additional 30 min incubation at room temperature (25 ˚C). The artificial cells were observed by confocal microscopy (Nikon A1R system).

**Confocal microscopy observation**. Artificial cells were observed by Nikon confocal microscopy system A1R. For the observation using GFP-Spo, a 488 nm laser was used with the setting as HV(GaAsP): 35, offset: 0, Intensity: 3.5, scan size: 1024, and scan speed: 0.5 frame/sec (Pixcel Dwell: 0.97 u s). For the NADPH permeability test, a 561 nm laser was used with the setting as HV(GaAsP): 200, offset: 0, Intensity: 80, scan size: 1024, and scan speed: 0.5 frame/sec (Pixcel Dwell: 0.97 u s). For the CoA permeability test, a 488 nm laser was used with the setting as HV(GaAsP): 20, offset: 0, Intensity: 2.0, scan size: 1024, and scan speed: 0.5 frame/sec (Pixcel Dwell: 0.97 u s). In all cases, images were taken as a set with a differential interference contrast (DIC) image.

Statistics and reproducibility—Error bars indicate standard deviation of at least triplicate measurements. *t* test was performed by Excel and applied to show statistical significance using of at least triplicate measurements. Exact number of replicates are shown in Supplementary Data 1.

**Reporting summary**. Further information on research design is available in the Nature Research Reporting Summary linked to this article.

## Data availability

The authors declare that the data supporting the findings of this study are available within the article, its Supplementary information files, and upon reasonable request. Source data in figures are provided in Supplementary Data 1.

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

## Acknowledgements

The expression plasmids for Fab enzymes, ACP, and TesA were kindly supplied by Prof. Chaitan Khosla (Stanford Univ.). The expression plasmid for AccDA was kindly supplied by Prof. John E. Cronan (Univ. of Illinois). Dr. Takashi Kanamori (GeneFrontier) supplied the PURE system. We thank Dr. Yoshi Kawai (Newcastle Univ.), Dr. Taku Oshima (Toyama Pref. Univ.), and Dr. Ken Takai (JAMSTEC) for valuable discussion, and also thank Mr. Gaku Sato and Dr. Shigeru Shimamura for assisting experiments. This work was supported by the Human Frontier Science Program (RPG0029/2020 to Y.K.), JST PRESTO (JPMJPR18K5 to Y.K.), JSPS KAKENHI (16H06156, 16KK0161, 16H00797, 26119704, 21H05156 to Y.K.), Astrobiology Center Project of the National Institutes of Natural Sciences (AB291017 to Y.K.), and an internal Grant-in-Aid from Earth-Life Science Institute (to Y.K.).

## Author contributions

S.E., R.M., and Y.S. performed most of the experiments, and M.F. and S.B. assisted some basic experiments. T.K. performed the mass spectrometry analysis and interpreted the data. Y.K. designed the framework of the research, analyzed the data, and wrote the paper. All authors discussed the results and commented on the paper.

## Competing interests

Y.K. is applying for a domestic patent (Japan) regarding the technology related to this work. The remaining authors declare no competing interests.
