## [Peer Review File · Communications Biology]

Reviewers' comments:

Reviewer #1 (Remarks to the Author):

The manuscript by Eto and colleagues described a pathway for synthesizing phospholipids inside synthetic minimal cells. This is very interesting work, addressing one of the largest unmet needs in the process of engineering synthetic minimal cells capable of self-replication.

The paper is well written, experiments are clear, rationale for all experiments is well described and experimental details are sufficient.

This is a complex pathway requiring concurrent expression of several genes. It would be crucial for interpretation of the results to know what percentage of the liposomes express the whole pathway, and what percentage of liposomes end up with fatty acid and phospholipid synthesis. Another crucial question to answer is whether and by how much the liposomes grow.

I think flow cytometry experiment would answer those questions.

Authors show molar yield of PA synthesis. This would be more informative if it was correlated with the mol% of the liposome membrane (150 μ M synthesized PA looks much more impressive in a sample of total of 0.5mM liposomes than in sample of 10mM liposomes). Basically, what was the yield as a percent of available substrates, proteins, and most importantly what % of liposome lipids came from the internal PA synthesis. That question is crucial to understanding how those results translate into practical problem of synthetic cell being able to self-renew it's lipids.

Authors use a droplet transfer method for liposome formation. This is indeed the most efficient method for formation of large unilamellar liposomes encapsulating biological cargo. There is, however, a known problem with mineral carryover in membranes formed that way.

It would be valuable for future work if Authors at least discuss the possibility of influence of residual mineral oil in the membrane on the liposome. Is it possible that presence of oil in the membrane affects the rate of incorporation of newly synthesized lipids? can it affect the activity of membrane associated proteins in the pathway?

Authors also mention, in materials and methods, use of thin film with extrusion for some liposome formation experiments. It is unclear which liposomes were formed using this method. I'm assuming that method was used for liposomes without PURE and DNA cargo, is that right?

Do newly synthesized lipids flip-flop?

The authors mention this issue only very briefly. Lack of morphological changes to liposomes might indicate flip-flop equilibration, but it also might indicate a whole range of compensation processes by the membrane (like other membrane lipids flip-flop instead, creating asymmetrical membranes, or the above-mentioned residual oil in the membrane affects membrane stability with new lipids accumulating only in the inner leaflet, just to name a few possibilities).

I think it would be valuable to show that the PAs of the type synthesized in this work can (or cannot) flip flop under the particular experimental conditions.

The NADPH permeability test is critical to characterization of the described system. The results presented on figure S16 are qualitatively convincing, but to fully estimate the efficiency and predict yields for lipid synthesis it would be much more valuable to have a quantitative measurement of permeability. Does NADPH equilibrate freely according to concentration gradient? what is the timeline of the permeability?

The microscopy assay used by Authors might be enough to answer those questions, if enough images are taken and analyzed by quantitative imaging processing software. Or some sort of size exclusion purification experiment might be good in this case, to get quantitative leakage data.

Minor points:

The paper is generally well written, but it would benefit from a pass of editing for clarity. Some sentences are awkward, for example "The liposome sample is generally stable for at least years, but it should be passed a bath sonication before use."

Figure 3D time course is missing a legend (an-d all shades of red are very similar to each other).

What is the DnaKJE chaperone mixture? it's mentioned briefly that it helps with solubility of PlsX, but it's unclear of what goes into that mix.

Reviewer #2 (Remarks to the Author):

Eto et. al. reported a cell-free phospholipid synthesis system that combines fatty acid synthesis and cell-free gene expression system synthesizing acyltransferases. Furthermore, phospholipid synthesis was conducted inside phospholipid membrane vesicles, which encapsulated all the components, and showed the phospholipids localized onto the mother membrane.

Phospholipid synthesis system is indispensable as the first step to reconstructing self-reproduction in artificial cells. This project is very challenging. This work provides a new way for a design of the self-reproducing artificial cells. I think this work will be interesting to a reasonable number of readers working in the field of artificial cells and phospholipid synthesis. Anyway, a major revision is required and the following questions should be addressed.

- 1, The author said "Our approach is a breakthrough for the construction of self-reproducing artificial cells in which membranes can sustainably grow and divide." Maybe the conclusions are exaggerated, since there are no data to support the division.
- 2, The permeability of NADH is greatly dependent on the membrane components. Please add a discussion on the possible mechanism.
- 3, Spo-GRP was used for the fluorescent imaging of FA. It is not clear on the recognition/ sensing mechanism. Can the method discriminate the new produced PA or PA from the mother vesicles?
- 4, In Figure 1A, it is better to have a reaction pathway to illustrate the cell-free phospholipid synthesis system, like Figure 4A, but without the lipid vesicles.
- 5, It is better to use the same style to prepare the scheme in figure 2A and figure 4A. Please use the same schematics to show the lipid, enzymes, and different components.
- 6, In figure 3B, the product of PA decreased as time, why? In Figure 3D, the substrates with different concentrations should be marked.
- 7, In figure 4A, the enzyme is suggested to mark in the schematic.
- 8, In figure 4B, a strong fluorescence was only observed for a vesicle in the center. It is suggested to supplement a new fluorescent imaging (small magnification) with many vesicles to show the PA production in all the vesicles.
- 9, Given that NADH is a co-enzyme factor to trigger the cascade reaction, it is better to investigate the PA production with and without addition of NADPH.
- 10, I am not sure if the approach will make the vesicle grow bigger, and it is suggested to supplement an experiment optical imaging of single vesicle to investigate their size evolution.
- 11, The phospholipid synthesis system can be conducted in water/oil emulsions for future work. It is likely that phospholipid production in w/o microreactor lead to a formation of lipid vesicles.

Reviewer #3 (Remarks to the Author):

The construction of self-reproducing artificial cells has been a research frontier in life science. Cell membrane is the key factor for cell growth and division. Phospholipid as the basic component of cell membrane, its in situ synthesis would be an important basis for building living artificial cells. This study performed phospholipid synthesis inside phospholipid membrane vesicles by cell-free reactions. It could be of interest, and as currently presented, there are still a few points that need to be addressed.

1. The authors need to give more explanations about the functions of eight FA-binding enzymes (FabA, FabB, FabD, FabF, FabG, FabH, FabI, and FabZ), acyl carrier protein (ACP), and thioesterase 1 (TesA).
2. Please give rational design reasons why focusing on specific enzymes for saturated and unsaturated FAs.
3. The solubility and activities of enzymes in this study are significant. Please give more information about them in the main text.
4. More discussions are needed to clarify the pros and cons of this study.

Responses to the Reviewers' comments

Reviewer #1

The manuscript by Eto and colleagues described a pathway for synthesizing phospholipids inside synthetic minimal cells. This is very interesting work, addressing one of the largest unmet needs in the process of engineering synthetic minimal cells capable of self-replication.

The paper is well written, experiments are clear, rationale for all experiments is well described and experimental details are sufficient.

We are very grateful for the positive assessment of reviewer 1.

This is a complex pathway requiring concurrent expression of several genes. It would be crucial for interpretation of the results to know what percentage of the liposomes express the whole pathway, and what percentage of liposomes end up with fatty acid and phospholipid synthesis. Another crucial question to answer is whether and by how much the liposomes grow.

I think flow cytometry experiment would answer those questions.

In the case of single protein synthesis inside giant vesicles, it has been reported that more than 95 % of the vesicles show successful protein (GFP) synthesis (Fig.2 in Nishimura *et al.* 2014 *RSC Advances*). However, different from single protein synthesis, it is technically difficult to determine what percentage of the artificial cells produce multi proteins in balanced concentrations. For this reason, we first checked the balanced protein syntheses (and the resulting phospholipid synthesis) *in vitro*, then encapsulated them into giant vesicles. Considering the data of the single protein synthesis, we think the ratio of successfully multi-synthesizing cells is similar to that of the single synthesis.

Measuring the GFP-Spo labeled artificial cells by a flow cytometer is a good idea, but, unfortunately, we do not have equipment in our laboratory and we cannot keep enough machine time for a common flow cytometer. Furthermore, most vesicles are adhering each other forming multicellular structures that interfere with obtaining a clear distribution pattern of the cells. Instead of analyzing with a flow cytometer, we counted the number of the labeled cells on

the microscopy images and calculated their ratio. Approximately 10-20 % (fluorescent cells/total cells*100) of the artificial cells showed significant fluorescence, which are assumed completely performed the reaction from the multi-protein syntheses to PA synthesis.

Regarding this, we added the following sentence in the paragraph “**Design and construction of artificial cells synthesizing phospholipids**” in the Result section.

“By counting the labeled vesicles, we estimated that approximately 10-20 % of the artificial cells show significant fluorescence that was assumed to have completely performed the reaction from the acyltransferase syntheses to PA synthesis.”

About the growth of the reacted artificial cells, we did not mention it in this study because we cannot remove the risk of artifact in the current condition. The current system synthesizing phospholipids from acetic acid by recycling CoA has the ability to produce max. 100 μ M phospholipids. This corresponds to 10 % of the phospholipids of the mother vesicle (dia. 30 μ m). It is very difficult to distinguish the difference of 10% of the surface area. More seriously, lipid membrane vesicles easily change their shape just by lipid dynamics, e.g. osmotic pressure difference, etc. Therefore, we decided to not discuss the growth of the artificial cells.

Currently, we are working to overcome this problem and will soon determine the direct reason that is limiting productivity. When we could improve the productivity of the system and get a significant increase in the surface area, we will be able to clearly show the growth of the vesicles.

Authors show molar yield of PA synthesis. This would be more informative if it was correlated with the mol% of the liposome membrane (150 μ M synthesized PA looks much more impressive in a sample of total of 0.5mM liposomes than in sample of 10mM liposomes). Basically, what was the yield as a percent of available substrates, proteins, and most importantly what % of liposome lipids came from the internal PA synthesis. That question is crucial to understanding how those results translate into practical problem of synthetic cell being able to self-renew it's lipids.

In this study, we performed phospholipid synthesis in solution (in vitro) for the data of Fig.1-3, then performed it inside vesicles for the data of Fig. 4. When we

performed PA synthesis in vitro, we supplied small-size liposomes (ca. 200nm (dia.)) in the reaction mixture. The liposomes have the role of a localization place for the synthesized PlsXYC and the synthesized lipids. The liposome concentration in the reaction mixture was 1.3 nM, which consist of 2.1 mM phospholipids. Therefore, when 150 uM PA was synthesized, this corresponds to 7 % of the total phospholipids forming the liposomes.

Following the suggestion by Reviewer 1, we added the sentence explaining this thing in the paragraph of **“Phospholipid synthesis by coupling fatty acid synthesis and acyltransferases”** in the Result section, as follows.

“In this condition, 2.1 mM phospholipids, which are forming 1.3 nM liposomes, were added from the beginning, thus the synthesized 400 μM PA corresponds to approximately 20 % of the liposome-forming lipids.”

On the other hand, when 100 uM PA was synthesized inside the artificial cell (30 um (dia.)), this corresponds to 10 % of the surface area of the artificial cell membrane as shown in Table S1. Therefore, the answer to the question “what % of liposome lipids came from the internal PA synthesis” is 10%. This has been described in the paragraph **“Design and construction of artificial cells synthesizing phospholipids”** in the Result section as *“..., it corresponds to 10 mol% of the phospholipids forming the artificial cell membrane (in the case of 30 μM diameter)”*.

Turnover rates of the enzymes are an important aspect of this system. We have tried to identify them but it is not straightforward since the system is composed of a multi-reaction network. For example, from the data of Fig. 1C or D, we can estimate 19-26 uM fatty acids (C12-C18) are synthesized per min in the initial velocity. However, the fatty acid synthesis system consists of a mixture of 9 kinds of Fab enzymes and the obtained fatty acids are the consequence of multiple rounds of hydrocarbon chain elongation, thus calculating the turnover rate in a typical way is difficult. Additionally, this fatty acid synthesis system also includes TesA which was not used in the reaction that was performed inside vesicles.

To calculate the turnover of the cell-free synthesized acyltransferases, PlsXYC, is also complicated because the amount of those substrates depends on the flux of the upstream reaction (FA synthesis rate). Therefore, we cannot satisfy

the premise that there is a sufficient amount of substrate required for kinetic analysis.

Contrary, we can perform kinetic analysis for acetyl-CoA synthetase (ACS) and acetyl-CoA carboxylases (AccDA). Following the reviewer's suggestions, we determined the initial velocities of ACS and AccDA as 425 ± 79 and 300 ± 65 $\mu\text{mol}/\text{min}/\text{mg}$ respectively. This means that the turnover rate of ACS and AccDA are 496 ± 92 and 333 ± 72 s^{-1} , respectively. Although the obtained data shows that the velocities of both enzymes are similar, the carboxylation reaction of AccDA resulted in an equilibrium state before depleting the substrate. This fact implies that the acetyl-CoA carboxylation step could be one of the limiting steps for the overall reaction rate downstream.

This thing was added in the Result section as *“The specific activities of ACS and AccDA (in the presence of $10 \mu\text{M}$ AccBC) were 425 ± 79 and $300 \pm 65 \mu\text{mol}/\text{min}/\text{mg}$, respectively.”*, and in the Discussion section as *“In fact, the malonyl-CoA synthesis rate of the acetyl-CoA carboxylase reached to equilibrium state before depleting the substrate, in contrast to the acetyl-CoA synthesis rate of the ACS that lasted to the maximum yield.”*. Additionally, the kinetics data was newly added as Fig. S10. To answer the suggested point “... practical problem of synthetic cell being able to self-renew it's lipids”, we have described in the Discussion section as *“However, there is a rate-limiting step in the malonyl-CoA synthesis, which is probably regulated by the acyl-ACP intermediates³⁸. To increase the productivity of lipid synthesis, it may be necessary to replace the acetyl-CoA carboxylase with an enzyme derived from another species that can circumvent the regulation by acyl-ACP intermediates.”*

Authors use a droplet transfer method for liposome formation. This is indeed the most efficient method for formation of large unilamellar liposomes encapsulating biological cargo. There is, however, a known problem with mineral carryover in membranes formed that way.

It would be valuable for future work if Authors at least discuss the possibility of influence of residual mineral oil in the membrane on the liposome. Is it possible that presence of oil in the membrane affects the rate of incorporation of newly synthesized lipids? can it affect the activity of membrane associated proteins in the pathway?

Authors also mention, in materials and methods, use of thin film with extrusion for some liposome formation experiments. It is unclear which liposomes were

formed using this method. I'm assuming that method was used for liposomes without PURE and DNA cargo, is that right?

As suggested by Reviewer 1, it has been known there is residual oil between the leaflets. However, it seems that the residual oil localizes at the north pole of the vesicle rather than equally distributes within the bilayer. This is because the oil density is lighter than water. For this reason, we consider that the effect of the residual oil is only local. Contrary, the distribution of membrane proteins is dynamic on a lipid membrane due to the fluid mosaic model. Taking all, the effect of residual oil on membrane protein may be limited. The same theory could be applied to the membrane insertion of newly synthesized lipids that will pass through the membrane proteins (PIsXYC).

We thank to the suggestion by Reviewer1 and added the following sentences in the Discussion section.

“Another point, it has been known that there is a certain amount of residual oil between the leaflets of giant vesicle membrane that was formed by the droplet transfer method. Although we cannot eliminate the possibility of the residual oil in our artificial cell membrane, we think that the effect of the oil on activities of the membrane-associated proteins is not significant, because the oil localizes at the north pole of the vesicle rather than equally distributes within the lipid bilayer. This is because the oil density is lighter than water. Contrary, membrane-associated proteins are dynamic on a lipid membrane due to the fluid mosaic model. Therefore, we think that the effect of the residual oil on membrane-associated proteins may be limited.”

As described above, liposomes were supplied into the PURE system for performing in vitro protein syntheses and phospholipid synthesis. In this condition, synthesized proteins (PIsXYC) localize onto the liposome membrane. Then, while phospholipid synthesis, the synthesized phospholipids insert into the same liposome membrane.

We added the following sentence in the Materials and Methods section.

“Small size liposomes were used for in vitro phospholipid synthesis reaction.”

Do newly synthesized lipids flip-flop?

The authors mention this issue only very briefly. Lack of morphological changes to liposomes might indicate flip-flop equilibration, but it also might indicate a whole range of compensation processes by the membrane (like other

membrane lipids flip-flop instead, creating asymmetrical membranes, or the above-mentioned residual oil in the membrane affects membrane stability with new lipids accumulating only in the inner leaflet, just to name a few possibilities).

I think it would be valuable to show that the PAs of the type synthesized in this work can (or cannot) flip flop under the particular experimental conditions.

Since we have detected the presence of PA on the outer leaflet of the artificial cell membrane, it is thought that a certain amount of PA was flip-flopped in the particular experimental condition. We are not sure about the direct reason why the artificial cells do not change their shape, but we think that insufficient phospholipid productivity is also one of the possible reasons. The question raised by Reviewer 1 is important regarding the creation of growing artificial cells. We are currently studying the speed of the flip-flop of the inner layer PA by making asymmetric membrane vesicles and monitoring the outer layer PA, but this is another project. We think we can show some results about this in the next paper.

We added the following sentence in the Discussion section.

“A more detailed analysis of the physicochemical property of the membrane will be needed to clarify how the flip-flops of lipid in the growing membrane affect the shape change of vesicles.”

The NADPH permeability test is critical to characterization of the described system. The results presented on figure S16 are qualitatively convincing, but to fully estimate the efficiency and predict yields for lipid synthesis it would be much more valuable to have a quantitative measurement of permeability. Does NADPH equilibrate freely according to concentration gradient? what is the timeline of the permeability?

The microscopy assay used by Authors might be enough to answer those questions, if enough images are taken and analyzed by quantitative imaging processing software. Or some sort of size exclusion purification experiment might be good in this case, to get quantitative leakage data.

Following the suggestion by Reviewer 1, we tried to obtain quantitative data to know the speed of membrane permeability of NADPH. However, as a result, we could not take reliable data with the current approach. In order to monitor the

presence of NADPH, we have used a fluorescent probe of the commercial kit (<https://www.cellbiolabs.com/sites/default/files/MET-5031-nadp-nadph-assay.pdf>). This system is composed of two kinds of reagents, “electron mediator” stimulating fluorescent the fluorometric probe with NADPH and “cycling enzyme” charging electron to NADP+, but we used only “electron mediator” reagent to monitor the permeated NADPH. The problem is this probe takes time to completely react with NADPH. We checked the time to arrive at the endpoint of fluorescent growing. As shown in the following data, it takes about 30 min in the given condition. Actually, the manufacturer’s protocol of the kit instructs to incubate the sample for 1-2 hours at room temperature. So that there is a time lag between binding with NADPH and monitoring the fluorescence. Since we aim to measure the initial speed of the membrane permeability of NADPH, the use of this probe is not suited for our purpose. To perform the accurate measuring of NADPH permeability, we may need other approaches such as the use of Confocal Raman Microscopy, etc., that we cannot access immediately. Therefore, although the rate of NADPH permeability is an important matter, we would like to work on it as the next project.

Fluorescence growing of NADPH probe. 8 mM NADPH and Fluorometric probe were mixed and fluorescence intensities were measured each 5 minutes by a plate reader.

Minor points:

The paper is generally well written, but it would benefit from a pass of editing for clarity. Some sentences are awkward, for example “The liposome sample is generally stable for at least years, but it should be passed a bath sonication before use.”

We changed the suggested sentence as below.

“The liposome sample is generally stable for at least years, but it should be treated by a bath sonication before use.”

We also scanned the manuscript and changed some awkward sentences.

Figure 3D time course is missing a legend (and all shades of red are very similar to each other).

We corrected Figure 3D and changed the color.

What is the DnaKJE chaperone mixture? it's mentioned briefly that it helps with solubility of PlsX, but it's unclear of what goes into that mix.

We added the following sentence in the paragraph of **“Design of artificial phospholipid synthesis system combining *in vitro* fatty acid synthesis and cell-free system”** in the Result section.

“The DnaKJE is the mixture of DnaK, DnaJ, and GrpE chaperons of E. coli.”

Reviewer #2 (Remarks to the Author):

Eto et. al. reported a cell-free phospholipid synthesis system that combines fatty acid synthesis and cell-free gene expression system synthesizing acyltransferases. Furthermore, phospholipid synthesis was conducted inside phospholipid membrane vesicles, which encapsulated all the components, and showed the phospholipids localized onto the mother membrane. Phospholipid synthesis system is indispensable as the first step to reconstructing self-reproduction in artificial cells. This project is very challenging. This work provides a new way for a design of the self-reproducing artificial cells. I think this work will be interesting to a reasonable number of readers working in the field of artificial cells and phospholipid synthesis. Anyway, a major revision is required and the following questions should be addressed.

We are grateful for the positive assessment of Reviewer 2.

1, The author said “Our approach is a breakthrough for the construction of self-reproducing artificial cells in which membranes can sustainably grow and

divide.” Maybe the conclusions are exaggerated, since there are no data to support the division.

We changed the sentence as follows.

“Our approach would be a platform for the construction of self-reproducing artificial cells since the membrane can grow sustainably.”

2, The permeability of NADH is greatly dependent on the membrane components. Please add a discussion on the possible mechanism.

We added the following sentences in the Discussion section as follows.

“Importantly, NADPH permeates the vesicle membrane consisting of 20 mol% POPG. Although the detailed mechanism is unknown, maybe NADPH interacts with the membrane and forms transient pores by defecting the stability of the membrane.”

3, Spo-GRP was used for the fluorescent imaging of FA. It is not clear on the recognition/ sensing mechanism. Can the method discriminate the new produced PA or PA from the mother vesicles?

To explain the mechanism of GFP-Spo fusion protein specifically binds to PA, we modified the following sentence and added a citation in the section **“Design and construction of artificial cells synthesizing phospholipids”**.

“The GFP-Spo is a fusion protein of GFP and Spo20, which contains an amphipathic motif that specifically recognizes PA on the membrane environment³³.”

Actually, we have missed giving citations about the Spo protein. Thank you for the suggestion.

PA is not included in the mother vesicle. All PAs were synthesized inside and localized onto the mother vesicle membrane.

4, In Figure 1A, it is better to have a reaction pathway to illustrate the cell-free phospholipid synthesis system, like Figure 4A, but without the lipid vesicles.

As suggested, we modified Figure 1A showing the reaction pathway in the cell-free system. The related figure legend was also modified.

5, It is better to use the same style to prepare the scheme in figure 2A and figure 4A. Please use the same schematics to show the lipid, enzymes, and different components.

As pointed out by reviewer 2, we changed the style of Figure 2A to match that of Figure 4A. However, we did not change the style of Acyl-ACP because we need to emphasize that nascent fatty acid elongates in the cavity of ACP protein.

6, In figure 3B, the product of PA decreased as time, why? In Figure 3D, the substrates with different concentrations should be marked.

About the decrease of the PA product, we do not understand the exact reason. Since there is a report showing some kind of acyltransferase has reverse activity (Yamashita et. Al. 2003, JBC), it could be due to the reverse activity of the cell-free synthesized acyltransferase. We added the following sentences in the paragraph “**Recycling of CoA towards sustainable phospholipid synthesis**” in the Result section.

“We also found a decrease in PA products after 10 minutes when 4 or 8 mM malonyl-CoA was supplied. Although the reason is unknown, this is probably due to the reverse activity of the acyltransferase.”

We modified Figure 3D as suggested by Reviewer 2.

7, In figure 4A, the enzyme is suggested to mark in the schematic.

We modified Figure 4A as suggested and replaced it.

8, In figure 4B, a strong fluorescence was only observed for a vesicle in the center. It is suggested to supplement a new fluorescent imaging (small magnification) with many vesicles to show the PA production in all the vesicles.

We have shown an image with a small magnification including many vesicles in Fig. S17.

To emphasize this, we added the following sentence in the same paragraph.

“We also confirmed the successful PA-synthesizing artificial cells in small magnification images with many vesicles (Fig. S17).”

9, Given that NADH is a co-enzyme factor to trigger the cascade reaction, it is better to investigate the PA production with and without addition of NADPH.

PA production with the addition of NADPH is shown in Figure 4B (the top line). About “without addition of NADPH”, we did the experiment and newly added the result in Figure 4B. The legend was modified as *“The artificial cells missing CoA, DNA, or NADPH were also reacted for the controls.”*

This was mentioned in the Result section as *“The synthesis of PA was further confirmed by omitting the template DNAs of acyltransferases or the addition of NADPH.”*

10, I am not sure if the approach will make the vesicle grow bigger, and it is suggested to supplement an experiment optical imaging of single vesicle to investigate their size evolution.

As suggested by Reviewer2, observation of the size evolution of the vesicles is one of the main goals of this study. However, the current system that synthesizes phospholipids from acetic acid by recycling CoA has the ability to produce only 100 μM phospholipids. This corresponds to 10 % of the phospholipids of the mother vesicle (dia. 30 μm). It is very difficult to distinguish

the growth of 10% surface area by microscopic observation. More seriously, lipid membrane vesicles easily change their shape just by lipid dynamics. Therefore, we think that we should be very careful to show the morphological change of the artificial cells to avoid artifacts. For this reason, we did not mention the shape change at this time. Currently, we are working to overcome this problem and will soon determine the direct reason limiting the productivity. When further productivity was achieved (e.g. >50% of the surface area), we will be able to clearly show the size change of the vesicles.

11, The phospholipid synthesis system can be conducted in water/oil emulsions for future work. It is likely that phospholipid production in w/o microreactor lead to a formation of lipid vesicles.

It is very interesting to conduct our system in water/oil emulsions in the future. As suggested by reviewer2, there is a chance to form lipid vesicles within the emulsions. However, before achieving this reaction, we need to solve the problem that NADPH inhibits the protein (acyltransferases) synthesis of the cell-free system. Although we do not know the direct reason why NADPH blocks the reaction, it could be solved by coupling with an NADP⁺ reduction system using a low concentration of NADPH.

We thank his/her valuable suggestion.

Reviewer #3 (Remarks to the Author):

The construction of self-reproducing artificial cells has been a research frontier in life science. Cell membrane is the key factor for cell growth and division. Phospholipid as the basic component of cell membrane, its in situ synthesis would be an important basis for building living artificial cells. This study performed phospholipid synthesis inside phospholipid membrane vesicles by cell-free reactions. It could be of interest, and as currently presented, there are still a few points that need to be addressed.

We thank Reviewer 3 for the fair assessment of our work.

1. The authors need to give more explanations about the functions of eight FA-

binding enzymes (FabA, FabB, FabD, FabF, FabG, FabH, FabI, and FabZ), acyl carrier protein (ACP), and thioesterase 1 (TesA).

We added the following sentences in the first paragraph of the Result section.

“Briefly, FabA and FabZ are 3-hydroxyacyl-ACP dehydrases (Fig. S2). FabB, FabF, and FabH are 3-ketoacyl-ACP synthase I, II, and III, respectively. FabD is malonyl-CoA:ACP transacylase. FabG and FabI are 3-ketoacyl-ACP reductase and enoyl-ACP reductase, respectively. ACP is a carrier of the growing fatty acid chain in fatty acid biosynthesis. TesA is a thioesterase specific for fatty acid thioesters.”

2. Please give rational design reasons why focusing on specific enzymes for saturated and unsaturated FAs.

We added the following sentence in the first paragraph of the Result section.

“The balance between saturated and unsaturated fatty acids within the structure of phospholipids is important in terms of the fluidity of lipid membrane.”

3. The solubility and activities of enzymes in this study are significant. Please give more information about them in the main text.

We added the following sentence in the last paragraph of the Result section.

“In this system, Fab enzymes, ACP, ACS, and ACCs are encapsulated as soluble recombinant proteins that exhibit the activities within the vesicle lumen. On the other hand, PlsX, PlsY, and PlsC are synthesized from the genes within the vesicle lumen because these are membrane-localized proteins. “

4. More discussions are needed to clarify the pros and cons of this study.

We added the following sentence in the Discussion section to emphasize the pros of our artificial cell system. We did not add further cons because those have been already described in the original text.

“Another advantage is we can produce phospholipids avoiding the accumulation of the by-product by reusing the used CoA and by recruiting acetic acid as a carbon source.”

REVIEWERS' COMMENTS:

Reviewer #1 (Remarks to the Author):

The Authors addressed all my comments and answered all my questions. Thank you.

Reviewer #2 (Remarks to the Author):

The questions have been addressed well and I don't have any new comments. I suggest this paper for acceptance.

Reviewer #3 (Remarks to the Author):

The authors have addressed all my concerns. Suggest the acceptance.